# A Surrogate Objective Framework for Prediction+Optimization with Soft Constraints

**Kai Yan**[*]
Department of Computer Science
University of Illinois at Urbana-Champaign
Urbana, IL 61801
kaiyan3@illinois.edu

**Jie Yan**
Microsoft Research
Beijing, China
jiey@microsoft.com

**Chuan Luo**
Microsoft Research
Beijing, china
chuan.luo@microsoft.com

**Liting Chen**[*]
Microsoft Research
Beijing, China
98chenliting@gmail.com

**Qingwei Lin**[†]
Microsoft Research
Beijing, China
qlin@microsoft.com

**Dongmei Zhang**
Microsoft Research
Beijing, China
dongmeiz@microsoft.com

## Abstract

Prediction+optimization is a common real-world paradigm where we have to predict problem parameters before solving the optimization problem. However, the criteria by which the prediction model is trained are often inconsistent with the goal of the downstream optimization problem. Recently, decision-focused prediction approaches, such as SPO+ and direct optimization, have been proposed to fill this gap. However, they cannot directly handle the soft constraints with the $max$ operator required in many real-world objectives. This paper proposes a novel analytically differentiable surrogate objective framework for real-world linear and semi-definite negative quadratic programming problems with soft linear and non-negative hard constraints. This framework gives the theoretical bounds on constraints' multipliers, and derives the closed-form solution with respect to predictive parameters and thus gradients for any variable in the problem. We evaluate our method in three applications extended with soft constraints: synthetic linear programming, portfolio optimization, and resource provisioning, demonstrating that our method outperforms traditional two-staged methods and other decision-focused approaches.

## 1 Introduction

Mathematical optimization (a.k.a. mathematical programming), e.g., linear and quadratic programming, has been widely applied in decision-making processes, such as resource scheduling [1], goods production planning [2], portfolio optimization [3], and power scheduling [4]. In practice, problem parameters (e.g., goods demands, and equity returns) are often contextual and predicted by models with observed features (e.g., history time series). With the popularity of machine learning techniques and increasing available data, prediction+optimization has become a normal paradigm [5].

---

[*]Contributed during internship at Microsoft Research. † Corresponding Author.

35th Conference on Neural Information Processing Systems (NeurIPS 2021).

Prediction becomes critical to the performance of the full prediction+optimization workflow since modern optimization solvers (e.g., Gurobi [6] and CPLEX [7]) can already efficiently find optimal solutions for most large scale optimization problems. Traditionally, prediction is treated separately as a general supervised learning problem and learned through minimizing a generic loss function (e.g., mean squared error for regression). However, studies have shown that minimization of the fitting errors does not necessarily lead to better final decision performance [8, 9, 10, 11].

Recently, a lot of efforts on using optimization objective to guide the learning of prediction models have been made, which are *decision-focused* when training prediction models instead of using traditional prediction metrics, e.g. mean squared error losses. For linear objectives, the 'Smart Predict then Optimize' ([10]) proposes the SPO+ loss function to measure the prediction errors against optimization objectives, while direct optimization ([12]) updates the prediction model's parameters by perturbation. For quadratic objectives, OptNet [13, 14] implements the optimization as an implicit layer whose gradients can be computed by differentiating the KKT conditions and then back propagates to prediction neural network. CVXPY [15] uses similar technologies with OptNet but extends to more general cases of convex optimization. However, all the above state-of-the-art approaches do not contain the soft constraints in their objectives. In this paper, We consider linear 'soft-constraints', a penalty in the form of $\max(z, 0)$, where $z = Cx - d$ is a projection of decision variable $x \in \mathbb{R}^n$ and context variables $C \in \mathbb{R}^{m \times n}, d \in \mathbb{R}^m$. Such soft constraints are often required in practice: for example, they could be the waste of provisioned resources over demands or the extra tax paid when violating regulatory rules. Unfortunately, the $\max(\cdot, 0)$ operator is not differentiable and thus cannot be directly handled by these existing approaches. To differentiate soft constraints is a primary motivation of this paper.

In this paper, we derive a surrogate objective framework for a broad set of real-world linear and quadratic programming problems with linear soft constraints and implement decision-focused differentiable predictions, with the assumption of non-negativity of hard constraint parameters. The framework consists of three steps: 1) rewriting all hard constraints into piece-wise linear soft constraints with a bounded penalty multiplier; 2) using a differentiable element-wise surrogate to substitute the piece-wise objective, and solving the original function numerically to decide which segment the optimal point is on; 3) analytically solving the local surrogate and obtaining the gradient; the gradient is identical to that of the piecewise surrogate since the surrogate is convex/concave such that the optimal point is unique.

Our main contributions are summarized as follows. First, we propose a differentiable surrogate objective function that incorporates both soft and hard constraints. As the foundation of our methodology, in Section 3, we prove that, with reasonable assumptions generally satisfied in the real world, for linear and semi-definite negative quadratic programming problems, the constraints can be transformed into soft constraints; then we propose an analytically differentiable surrogate function for the soft constraints $\max(\cdot, 0)$. Second, we present the derived analytical and closed-form solutions for three representative optimization problems extended with soft constraints in Section 4 – linear programming with soft constraints, quadratic programming with soft constraints, and asymmetric soft constraint minimization. Unlike KKT-based differentiation methods, our method makes the calculation of gradients straightforward for predicting context parameters in any part of the problem. Finally, we apply with theoretical derivations and evaluate our approach in three scenarios, including synthetic linear programming, portfolio optimization, and resource provisioning in Section 4, empirically demonstrate that our method outperforms two-stage and other predict+optimization approaches.

## 2 Preliminaries

### 2.1 Real-world Optimization Problems with Soft Constraints

Our target is to solve the broad set of real-world mathematical optimization problems extended with a $max(z, 0)$ term in their objectives where $z$ depends on decision variables and predicted context parameters. In practice, $max(z, 0)$ is very common; for example, it may model overhead of under-provisioning, over-provisioning of goods, and penalty of soft regulation violations in investment portfolios. We call the above $max(z, 0)$ term in an objective as *soft constraints*, where $z \leq 0$ is allowed to violate as long as the objective improves.

The general problem formulation is

$$\max_x g(\theta, x) - \alpha^T \max(z, 0),\ z = Cx - d,\ \text{s.t. } Ax \le b, Bx = c,\ x \ge 0 \tag{1}$$

where $g$ is a utility function, $x \in \mathbb{R}^n$ the decision variable, and $\theta \in \mathbb{R}^n$ the predicted parameters.

Based on observations on a broad set of practical problem settings, we impose two assumptions on the formulation, which serves as the basis of following derivations in this paper. First, we assume $A \in \mathbb{R}^{m_1 \times n} \ge 0$, $b \in \mathbb{R}^{m_1} \ge 0$, $B \in \mathbb{R}^{m_2 \times n} \ge 0$, and $c \in \mathbb{R}^{m_2} \ge 0$ hold. This is because for problems with constraint on weights, quantities or their thresholds, these parameters are naturally non-negative. Second, we assume the linearity of soft constraints, that is $z = Cx - d$, where $z \in \mathbb{R}^{m_3}$, $C \in \mathbb{R}^{m_3 \times n}$, and $d \in \mathbb{R}^{m_3}$. This form of soft constraints make sense in wide application situations when describing the penalty of goods under-provisioning or over-provisioning, a vanish in marginal profits, or running out of current materials.

Now we look into three representative instances of Eq.1, extracted from real-world applications.

**Linear programming with soft constraints**, where $g(x, \theta) = \theta^T x$. The problem formulation is

$$\max_x \theta^T x - \alpha^T \max(Cx - d, 0), \text{s.t. } Ax \le b, Bx = c, x \ge 0 \tag{2}$$

where $\alpha \ge 0$. Consider the application of logistics where $\theta$ represents goods' prices, $\{A, B, C\}$ are the capability of transportation tools, and $\{b, c\}$ are the thresholds. Obviously, $A, b, B, c \ge 0$ hold.

**Quadratic programming with soft constraints**, where $g(\theta, x) = \theta^T x - x^T Q x$. One example is the classic minimum variance portfolio problem [16] with semi-definite positive covariance matrix $Q$ and expected return $\theta$ to be predicted. We extend it with soft constraints which, for example, may represent regulations on portions of equities for some fund types. Formally with $\alpha \ge 0$, we have:

$$\max_x \theta^T x - x^T Q x - \alpha^T \max(Cx - d, 0), \text{s.t. } Bx = c,\ x \ge 0. \tag{3}$$

**Optimization of asymmetric soft constraints.** This set of optimization problems have the objective to match some expected quantities by penalizing excess and deficiency with probably different weights. Such formulation represents widespread resource provisioning problems, e.g., power[4] and cloud resources[1], where we minimize the cost of under-provisioning and over-provisioning against demands. Formally with $\alpha_1, \alpha_2 > 0$, we have:

$$\max_x -(\alpha_1^T \max(Cx - d, 0) + \alpha_2^T \max(d - Cx, 0)),\ \text{s.t. } Bx = c,\ x \ge 0. \tag{4}$$

In this paper, we consider a challenging task where $C$ is a matrix to be predicted with known constants $d$. In reality, the $Cx - d$ term may represent the "wasted" part when satisfying the actual need of $d$.

## 2.2 Prediction+Optimization

For compactness we write Eq.1 as $\max_{x \in \mathcal{X}} f(x, \theta)$, where $f$ is the objective function and $\mathcal{X}$ is the feasible domain. The solver for $f$ is to solve $x^* = \text{argmax}_{x \in \mathcal{X}} f(x, \theta)$. With parameters $\theta$ known, Eq.2–4 can be solved by mature solvers like Gurobi [6] and CPLEX [7].

In prediction+optimization, $\theta$ is unknown and needs to be predicted from some observed features $\xi \in \Xi$. The prediction model is trained on a dataset $D = \{(\xi_i, \theta_i)\}_{i=1}^N$. In this paper, we consider the prediction model, namely $\Phi$, as a neural network parameterized with $\psi$. In traditional supervised learning, $\Phi_\psi$ is learned by minimizing a generic loss, e.g., L1 (Mean Absolute Error) or L2 (Mean Squared Error), which measures the expected distance between predictions and real values. However, such loss minimization is often inconsistent with the optimization objective $f$, especially when the prediction model is biased [11].

Instead, decision-focused learning directly optimizes $\psi$ with respect to the optimization objective $f$, that is, $\max_\psi \mathbf{E}_{(\xi, \theta) \sim D}[f(\hat{x}^*(\Phi_\psi(\xi)), \theta)]$. The full computational flow is illustrated in Fig.1. In the gradient-based learning, update of $\psi$'s gradient is $\frac{\partial r}{\partial \psi} = \frac{\partial \hat{\theta}}{\partial \psi} \frac{\partial \hat{x}^*}{\partial \hat{\theta}} \frac{\partial r}{\partial \hat{x}^*}$, where utility $r = f(\hat{x}^*, \theta)$. The Jacobian $\partial \hat{\theta} / \partial \psi$ is computed implicitly by auto-differentiation of deep learning frameworks (e.g, PyTorch [17]), and $\partial r / \partial \hat{x}^*$ is analytical. The main challenge is to compute $\partial \hat{x}^* / \partial \hat{\theta}$, which depends on differentiating the $\text{argmax}$ operation. One recent approach is to rewrite the objective to be convex (by adding a quadratic regularizer if necessary), build and differentiate the optimality conditions (e.g.,

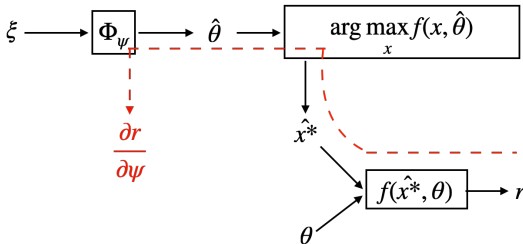

Figure 1: Computation graph of the decision-focused prediction methods.

KKT conditions) [18] which map $\hat{\theta}$ to the solution $\hat{x}^*$, and then apply implicit function theorem to obtain $\partial \hat{x}^* / \partial \hat{\theta}$. Alternatively, in this paper we propose a novel approach that rewrites the problem as an unconstrained problem with soft constraints and derives analytical solutions, based on our observations on the real-world problem structures and coefficient properties.

## 3   Methodology

Our main idea is to derive a surrogate function for $f$ with a closed-form solution such that the Jacobian $\frac{\partial x}{\partial \theta}$ is analytical, making the computation of gradient straightforward. Unlike other recent work [13, 14, 15, 3], our method does not need to solve KKT optimality condition system. Instead, by adding reasonable costs for infeasibility, we convert the constrained problem into an unconstrained one. With the assumption of concavity, we prove that there exist constant vectors $\beta_1, \beta_2, \beta_3$, such that Eq.1 can be equivalently transformed into an unconstrained problem:

$$\max_x L(x) = \max_x g(x, \theta) - \alpha^T \max(Cx - d, 0) - \beta_1^T \max(Ax - b, 0) - \beta_2^T |Bx - c| - \beta_3^T \max(-x, 0) \quad (5)$$

The structure of this section is as follows. Section 3.1 proves that the three types of hard constraints can be softened by deriving bounds of $\beta_1, \beta_2, \beta_3$; for this paper, we will assign each entry of the three vectors the equal value (with a slight abuse of notation, we denote $\beta_1 = \beta_2 = \beta_3 = \beta$ for the proofs of bounds; we align with the worst bound applicable to the problem formulation.) Section 3.2 proposes a novel surrogate function of $\max(\cdot, 0)$, such that the analytical form of $\frac{\partial x}{\partial \theta}$ can be easily derived via techniques of implicit differentiation [18] and matrix differential calculus [19] on equations derived by convexity [20]. Based on such derived $\frac{\partial x}{\partial \theta}$, we develop our end-to-end learning algorithm of prediction+optimization whose detailed procedure is described in Appendix C.

### 3.1   Softening the Hard Constraints

For any hard constraints $w = Ax - b \leq 0$, we denote its equivalent soft constraints as $H(w) = \beta^T \max(w, 0)$. $H(w)$ should satisfy two conditions: 1) for $w \leq 0$ (feasible $x$), $H(w) = 0$; 2) for $w \geq 0$ (infeasible $x$), $H(w)$ is larger than *the utility gain*(i.e., improvement of the objective value) $R = f(x, \theta) - \max_{x_1 : Ax_1 \leq b} f(x_1, \theta)$ by violating $Ax - b \leq 0$. Intuitively, the second condition requires a sufficiently large-valued $\beta > 0$ to ensure that the optimization on the unconstrained surrogate objective never violates the original $Ax \leq b$; to make this possible, we assume that the $l2$-norm of the derivative of the objective $f$ before conversion is bounded by constant $E$. The difficulty of requirement 2) is that the distance of a point to the convex hull $l$ is not bounded by the sum of distances between the point and each hyper-plane in general cases, so the utility gain obtained from violating constraints is unbounded. Fig. 2-(a) shows such an example which features the small angle between hyper-planes of the activated cone. We will refer such kind of 'unbounding' as "*acute angles*" below.

The main effort of this subsection is to analyze and bound the effect caused by such acute angles. Given a convex hull $\mathcal{C} = \{z \in \mathbb{R}^n | Az \leq b\}$ ($A \geq 0$ is not required here) and any point $x \notin \mathcal{C}$, let $x_0 \in \mathcal{C}$ be the nearest point of $\mathcal{C}$ to $x$, and $A'x \geq b'$ represent all active constraints at $x$, then all such active constraints must pass through $x_0$.[2] The domain $\mathcal{K} = \{z \in \mathbb{R}^n | A'z \geq b'\}$ is a cone or degraded

---

[2]Otherwise, there must exist an active constraint $i$ and a small neighbourhood of $x_0$, say $B(x_0, \epsilon)$, such that $\forall z \in B(x_0, \epsilon), A_i z - b_i \neq 0$, which implies that either constraint $i$ is inactive ($< 0$) or $x_0 \notin \mathcal{C}$ ($> 0$).

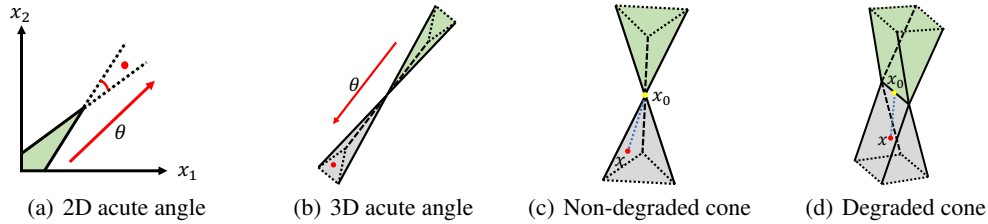



(a) 2D acute angle      (b) 3D acute angle      (c) Non-degraded cone      (d) Degraded cone



Figure 2: (a) and (b) are 2 and 3-dimensional "*acute angles*"; (c) and (d) shows two corresponding *activated cones* for given acute angles. The green area is the feasible region, $x$ is the red point and $x_0$ is the yellow point; the red $\theta$ is the derivative of an objective $g(x, \theta) = \theta^T x$.

cone where the tip of the cone is a subspace of $R^n$. For the rest of the paper, we will call $\mathcal{K}$ *activated cone*, as shown in Fig. 2. Note that for any degraded activated cone, $x - x_0$ is always perpendicular to the tip subspace; therefore, we may project the cone onto the complementary space of the tip and get the same multiplier bound on the projected activated cone with lower dimensions.

Ideally, we aim to eliminate the utility gain $R$ obtained from violating $A'x \leq b'$ with the penalty $\beta^T(A'x - b')$, *i.e.*, ensure $\beta^T(A'x - b') \geq R$ hold for any $x$. For the compactness of symbols and simplicity, we will assume that the main objective is $g(x, \theta) = \theta^T x$ in this section; however note that our proofs apply with the existence of soft constraints and quadratic terms in $g$, which is discussed at the beginning of Appendix A. With such assumption, we now give a crucial lemma, which is the core of our proof for most of our theorem:

**Lemma 1.** *(Bounding the utility gain) Let $R = f(x, \theta) - \max_{x_1 \in \mathcal{C}} f(x_1, \theta)$ be the utility gain, then $R \leq f(x, \theta) - f(x_0, \theta) \leq \frac{E}{\cos p_0} \sum_{i=1}^{n} (A_i'x - b_i')$, where $x$ is an infeasible point, $A'x \leq b'$ the active constraints at $x$, $p_0 = \angle(A_i'^{*}, \theta^*)$ where $A_i'^{*}$ and $\theta^*$ are the optimal solution of $\max_\theta \min_{A_i'} \cos\angle(A_i', \theta)$ (i.e., the maximin angle between $\theta$ and any hyperplane of the activated cone $\mathcal{K} = \{z \in \mathbb{R}^n | z - x_0 \in cone(A')\}$), and $x_0$ the projection of $x$ to the tip of cone $\mathcal{C} = \{z \in \mathbb{R}^n | A'z \leq b'\}$. $E$ is the upper bound of $||\theta||_2$. $\angle(\cdot, \cdot)$ denotes the angle of two vectors.*

Thus, $\frac{E}{\cos p_0}\mathbf{1}$ is a feasible choice of $\beta$, and it suffices by finding the lower bound for $\cos p_0$. For the rest of the section, we find the lower bound of $\cos p_0$ by exploiting the assumed properties of $A'$, *e.g.*, $A' \geq 0$; we give the explanation of the full proofs for all theoretical results in Appendix A.

### 3.1.1 Conversion of Inequality Constraints $Ax \leq b$

Let us first consider the constraints $w = Ax - b \leq 0$, where $A \geq 0$, $b \geq 0$. It is easy to prove that given $A \geq 0$ and $b \geq 0$, the distance of a point to the convex hull $Ax \leq b$ is bounded. More rigorously, we have the following theorem, which guarantees the feasibility of softening the hard constraints of inequalities $Ax \leq b$:

**Theorem 2.** *Assume the optimization objective $\theta^T x$ with constraints $Ax \leq b$, where $A \geq 0$, and $b \geq 0$. Then, the utility gain $R$ obtained from violating $Ax - b \leq 0$ has an upper bound of $O(\sum_i \max(w_i, 0)E)$, where $w = A'x - b'$, and $A'x \leq b'$ is the active constraints.*

### 3.1.2 Conversion of Inequality Constraints $x \geq 0$

With inequality constraints $Ax \leq b$ converted, we now further enforce $x \geq 0$ into soft constraints. It seems that the constraint of this type may form a very small angle to those in $w = Ax - b$. However, as $-x$ is aligned with the axes, we can augment $x \geq 0$ into soft constraints by proving the following theorem:

**Theorem 3.** *When there is at least one entry of $x \geq 0$ in the activated cone, the utility gain $R$ from deviating the feasible region is bounded by $O(\frac{n^{1.5}E\sum_i max(w_i, 0)}{\sin p})$, where $p$ is the smallest angle between axes and other constraint hyper-planes and $w$ is the union of $Ax - b$ and $-x$.*

Hence, we can set $\beta = O(\frac{n^{1.5}E}{\sin p})$. Specially, for binary constraints we may set $\beta = O(n^{1.5}E)$:



5



**Corollary 4.** *For binary constraints where the entries of $A$ are either $0$ or $1$, the utility gain $R$ of violating $x \geq 0$ constraint is bounded by $O(n^{1.5} E \sum_i \max(w_i, 0))$, where $w_i = A_i x - b_i$ or $-x$.*

which gives a better bound for a set of unweighted item selection problem (e.g. select at most $k$ items from a particular subset).

### 3.1.3  Conversion of Equality Constraints $Bx = c$

Finally, we convert $Bx = c$ into soft constraints. This is particularly difficult, as $Bx = c$ implies both $Bx \leq c$ and $-Bx \leq -c$, which will almost always cause acute angles. Let's first consider a special case where there is only one equality constraints and $A$ is an element matrix $I^{n \times n}$.

**Theorem 5.** *If there is only one equality constraint $Bx = c$ (i.e., $B$ degrades as a row vector, such like $\sum_i x_i = 1$) and special inequality constraints $x \geq 0$, $Ix \leq b$, then the utility gain $R$ from violating constraints is bounded by $O(\frac{n^{1.5} E \sum_i \max(w_i, 0)}{\sin p})$, where $p$ is the same with theorem 3, $w$ is the union of $Bx - c$ and $-x$.*

Intuitively, when there is only one equality constraint, the equality constraint can be viewed as an unequal one, for at most one side of the constraint can be in a non-degraded activated cone. Thus, we can directly apply the proof of Theorem 2 and 3, deriving the same bound $O(\frac{n^{1.5} E}{\sin p})$ for $\beta$.

Finally, we give bounds for general $Ax \leq b$, $Bx = c$ with $A, b, B, c \geq 0$ as below:

**Theorem 6.** *Given constraints $Ax \leq b$, $x \geq 0$, and $Bx = c$, where $A, B, b, c \geq 0$, the utility gain $R$ obtained from violating constraints is bounded by $O(\sqrt{n} \lambda_{max} \sum_i \max(w_i, 0))$, where $\lambda_{max}$ is the upper bound for eigenvalues of $P^T P$ ($P : x \to Px$ is an orthogonal transformation for an $n$-sized subset of normalized row vectors in $A, B$ and $-I$), and $w$ is the union of all active constraints from $Ax \leq b$, $x \geq 0$, $Bx \leq c$ and $-Bx \leq -c$.*

In this theorem, $P$ is generated by taking an arbitrary $n$-sized subset from the union of row vectors in $A, B$ and $-I$, orthogonizing the subset, and using the orthogonal transformation matrix as $P$; there are $\binom{n+m_1+m_2}{n}$ different cases of $P$, and $\lambda_{max}$ is the upper bound of eigenvalues of $P^T P$ over all possible cases of $P$. Note that there are no direct bounds on $\lambda_{max}$ with respect to $n$ and the angles between hyper-planes. However, empirical results (see Appendix A for details) show that for a diverse set of synthetic data distributions, $\lambda_{max} = O(n^2)$ follows. Therefore, empirically we can use a bound $O(\frac{n^{2.5} E}{\sin p})$ for $\beta$.[3] So far, we have proven that all hard constraints can be transformed into soft constraints with bounded multipliers. For compactness, Eq.5 is rewritten in a unified form:

$$L(x) = g(x, \theta) - \gamma^T \cdot max(C'x - d'), \text{ where } \gamma = \begin{bmatrix} \alpha \\ O(\frac{n^{2.5} E}{\sin p})\mathbf{1} \end{bmatrix}, \ C' = \begin{bmatrix} C \\ A \\ -B \\ B \\ -I \end{bmatrix}, \ d' = \begin{bmatrix} d \\ b \\ -c \\ c \\ 0 \end{bmatrix} \quad (6)$$

### 3.2  The Unconstrained Soft Constraints

As Eq.5 is non-differentiable for the max operator, we need to seek a relaxing surrogate for differentiation. The most apparent choice of the *derivative* of such surrogate $S(z)$ for $z = C'x - d'$ is sigmoidal functions; however, it is difficult to derive a closed-form solution for such functions, since sigmoidal functions yield $z$ in the denominator, and $z$ cannot be directly solved because $C$ is not invertible (referring to Appendix B for detailed reasons). Therefore, we have to consider a piecewise roundabout where we can first numerically solve the optimal point to determine which segment the optimal point is on, and then expand the segment to the whole space. To make this feasible, two assumptions must be made: 1) this function must be differentiable, and 2) the optimal point must be unique; to ensure this, the surrogate should be a convex/concave piece-wise function. The second property is for maintaining the optimal point upon segment expansion. Fortunately, there is one simple surrogate function satisfying our requirement:

$$S(z) = \begin{cases} 0 & \text{if } z < -\frac{1}{4K} \\ K(z + \frac{1}{4K})^2 & \text{if } -\frac{1}{4K} \leq z \leq \frac{1}{4K} \\ z & \text{if } z \geq \frac{1}{4K} \end{cases} \quad (7)$$

---

[3]For real optimization problems, we can sample $A'$ from $A$ and estimate the largest eigenvalue of $P^T P$.

Let $M$ and $U$ be diagonal matrices as the indicator of $S(z)$, which are $M_{i,i} = 2K[-\frac{1}{4K} \leq z_i \leq \frac{1}{4K}]$ and $U_{i,i} = [\frac{1}{4K} < z_i]$, where $[\cdot]$ is an indicator function. $K > 0$ is a hyper-parameter that needs to be balanced. Larger $K$ makes the function closer to original; however, if $K$ is too large, then the training process would be destabilized, because when the prediction error is large at the start of the training process, $\frac{\partial f}{\partial \theta}|_{\hat{x}}$ might be too steep. Then consider the optimal point for the unconstrained optimization problem maximizing $\theta^T x - \gamma^T \max(C'x - d', 0)$, by differentiaing on both sides, we can obtain:

$$\theta = C'^T M \text{diag}(\gamma)(C'x - d') + C'^T(\frac{1}{4K}M + U)\gamma \tag{8}$$

This equation reveals the closed-form solution of $x$ with respect to $\theta$, and thus the calculation of $\frac{\partial x}{\partial \theta}$ becomes straightforward:

$$\frac{\partial x}{\partial \theta} = (C'^T M \text{diag}(\gamma)C')^{-1} \tag{9}$$

$C'^T M \text{diag}(\gamma)C'$ is invertible as long as at least $n$ soft constraints are on the quadratic segment (*i.e.,* active), which is the necessary condition to fix a particular optimal point $n$ in $\mathcal{R}^n$. With such solution, we can train our prediction model with stochastic gradient descent (SGD). For each batch of problem instances, we first solve optimization problems numerically using solvers like Gurobi to get the matrix $M$ and $U$, and then calculate gradients with the analytical solution. The parameters of the prediction model are updated by such gradients. The sketch of our algorithm is outlined in Appendix C.

## 4 Applications and Experiments

We apply and evaluate our approach on the three problems described in Section 2, *i.e.,* linear programming, quadratic programming, and asymmetric soft constraint minimization. These problems are closely related to three applications respectively: synthetic linear programming, portfolio optimization, and resource provisioning, which are constructed using synthetic or real-world datasets. The detailed derivation of gradients for each application can be found in Appendix D.

In our experiments, the performance is measured in regret, which is the difference between the objective value when solving optimization over predicted parameters and the objective value when solving optimization over actual parameters. For each experiment, we choose two generic two-stage methods with $L1$-loss and $L2$-loss, as well as decision-focused methods for comparison baselines. We choose both SPO+[10] and DF proposed by Wilder *et al.* [9] for synthetic linear programming and DF only for portfolio optimization,[4] as the former is specially designed for for linear objective. For resource provisioning, we use a problem-specific weighted L1 loss, as both SPO+ and DF are not designed for gradients with respect to variables in the soft constraints. All reported results for each method are obtained by averaging on 15 independent runs with different random seeds.

As real-world data is more lenient than extreme cases, in practice we use a much lower empirical bound than the upper bound proved in section 3.1., e.g., constants of around 20 and $5\sqrt{n}$ where $n$ is the number of dimensions of decision variables. One rule of thumb is to start from a reasonable constant or a constant times $\sqrt{n}$, where such "reasonable constant" is the product of a constant factor (e.g. $5 - 10$) and a roughly estimated upper bound of $||\theta||_2$ (which corresponds to $E$ in our bounds) with specific problem settings; then alternately increase the constant and time an extra $\sqrt{n}$ while resetting the constant until the program stops diverging, and the hard constraints are satisfied. In our experiments, we hardly observe situations where such process goes for two or more steps.

### 4.1 Synthetic Linear Programming

**Problem setup.** The prediction dataset $\{(\xi_i, \theta_i)\}_{i=1}^N$ is generated by a general structural causal model ([21]), ensuring it is representative and consistent with physical process in nature. The programming parameters are generated for various scales in numbers of decision variables, hard constraints, and soft constraints. Full details are given in Appendix E.

---

[4]The method [3] and its variant with dimension reduction [9] have same performance in terms of regret on this problem, thus we choose the former for comparison convenience.

| | | Regret (the lower, the better) | | | | |
|---|---|---|---|---|---|---|
| $N$ | Problem Size | L1 | L2 | SPO+ [10] | DF [9] | Ours |
| 100 | (40, 40, 0) | 2.454±0.232 | 2.493±0.295 | 2.506±0.294 | 2.478±0.425 | **2.258±0.311** |
| | (40, 40, 20) | 2.626±0.307 | 2.664±0.303 | 2.667±0.281 | 2.536±0.376 | **2.350±0.263** |
| | (80, 80, 0) | 5.736±0.291 | 5.831±0.361 | 5.711±0.309 | 5.756±0.317 | **5.200±0.506** |
| | (80, 80, 40) | 4.786±0.403 | 4.786±0.596 | 4.939±0.382 | 4.902±0.537 | **4.570±0.390** |
| 1000 | (40, 40, 0) | 1.463±0.143 | 1.447±0.155 | 1.454±0.148 | 1.434±0.268 | **1.346±0.144** |
| | (40, 40, 20) | 1.626±0.141 | 1.613±0.110 | 1.618±0.103 | 1.529±0.151 | **1.506±0.102** |
| | (80, 80, 0) | 3.768±0.132 | 3.718±0.117 | 3.573±0.113 | 3.532±0.102 | **3.431±0.100** |
| | (80, 80, 40) | 2.982±0.176 | 2.913±0.172 | 2.879±0.148 | 3.351±0.212 | **2.781±0.165** |
| 5000 | (40, 40, 0) | 1.077±0.105 | 1.080±0.109 | 1.090±0.105 | 1.078±0.092 | **1.037±0.100** |
| | (40, 40, 20) | 1.283±0.070 | 1.277±0.077 | 1.298±0.077 | 1.291±0.091 | **1.220±0.071** |
| | (80, 80, 0) | 2.959±0.086 | 2.943±0.091 | 2.926±0.079 | 2.869±0.085 | **2.845±0.064** |
| | (80, 80, 40) | 2.239±0.122 | 2.224±0.106 | 2.234±0.122 | 2.748±0.165 | **2.172±0.098** |

Table 1: Performance comparison (regret mean with std. deviation) for the synthetic linear programming problem. $N$ is the size of the training dataset, and problem size is a triplet (# of decision variables' dimension, # of hard constraints, # of soft constraints).

**Experimental setup.** All five methods use the same prediction model – a fully connected neural network of two hidden layers with 128 neurons for each and ReLU [22] for activation. We use AdaGrad [23] as the optimizer, with learning rate 0.01 and gradient clipped at $1e-4$. We train each method for 40 epochs, and early stop when valid performance degrades for 4 consecutive epochs. Specially, to make DF[9] work right on the non-differentiable soft constraints, we first use black-box solvers to determine whether each soft constraint is active on the optimal point, and then optimize with its local expression (i.e. 2nd-order Taylor expansion at optimal point).

**Performance analysis.** Our experiments cover four programming problem scales with three prediction dataset sizes. Results are summarized in Table 1. In all cases, our method performs consistently better than two-stage methods, DF and SPO+. Even for the cases with only hard constraints (*i.e.,* the third parameter of problem size is 0), our method still has significant advantage, demonstrating its effectiveness on handling hard constraints. Surprisingly, although the main objective is linear, SPO+ often performs even worse than two-stage methods. Detailed analysis (see the appendix) shows that SPO+ quickly reaches the test optimality and then over-fits. This may be due to that, unlike our method, SPO+ loss is not designed to align with the soft constraints. This unawareness of soft constraint is also why DF is performing worse than our method, as DF is working on an optimization landscape that is non-differentiable at the boundary of soft constraints, on which the optimal point usually lies. Besides, with the increment of the samples in train data, the performance of all methods is improved significantly and the performance gap among ours and two-stage methods becomes narrow, which implies that prediction of two-stage methods becomes better and with lower biases. Even so, our method has better sample efficiency than two-stage methods.

We also investigated effect of the hyper-parameter $K$ in our surrogate max function, detailed in the appendix. Through our experiments, $K$'s effect to regret is not monotonic, and its optimal value varies for different problem sizes. Interestingly, $K$'s effect is approximately smooth. Thus, in practice, we use simple grid search to efficiently find the best setting of $K$.

## 4.2 Portfolio Optimization

**Problem and experimental setup.** The prediction dataset is daily price data of SP500 from 2004 to 2017 downloaded by Quandl API [24] with the same settings in [3]. We use the same fix as that in linear programming experiment to make DF[9] work with non-differentiable soft constraints, which was also used in [3] for non-convex optimization applications. Most settings are aligned with those in [3], including dataset configuration, prediction model, learning rate (0.01), optimizer (Adam), gradient clip (0.01) and number of training epochs (20). We set the number of soft constraints to 0.4 times of $n$, where $n$ is the number of candidate equities. For the soft constraint $\alpha^T \max(Cx - d, 0)$, $\alpha = \frac{15}{n}v$, where each element of $v$ is generated randomly at uniform from $(0, 1)$; the elements of matrix $C$ is generated independently from $\{0, 1\}$, where the probability of 0 is 0.9 and 1 is 0.1. $K$ is set as 100.

| #Equities | Regret measured in % (the lower, the better) | | | |
|---|---|---|---|---|
| | L1 | L2 | DF [9] | ours($K = 100$) |
| 50 | 4.426±0.386 | 4.472±0.385 | 4.016±0.389 | **3.662±0.238** |
| 100 | 4.262±0.231 | 4.320±0.229 | 3.500±0.252 | **3.214±0.138** |
| 150 | 3.878±0.281 | 3.950±0.287 | 3.419±0.281 | **3.109±0.162** |
| 200 | 3.755±0.236 | 3.822±0.273 | 3.406±0.287 | **3.152±0.183** |
| 250 | 3.721±0.205 | 3.751±0.212 | 3.335±0.175 | **3.212±0.135** |

Table 2: Performance comparison (regret mean with std. deviation) for portfolio optimization.

| $\alpha_1/\alpha_2$ | Regret (the lower, the better) | | | |
|---|---|---|---|---|
| | L1 | L2 | Weighted L1 | Ours($K = 0.05$) |
| 100 | 105.061±21.954 | 93.193±29.815 | 79.014±32.069 | **20.829±8.289** |
| 10 | 13.061±2.713 | 13.275±6.208 | 7.743±1.305 | **2.746±1.296** |
| 1 | **4.267±0.618** | 5.136±0.722 | **4.267±0.618** | 5.839±0.512 |
| 0.1 | 10.846±1.606 | 13.619±2.195 | 16.462±2.093 | **10.240±1.248** |
| 0.01 | 99.145±21.159 | 118.112±29.957 | 230.825±91.184 | **94.341±29.821** |

Table 3: Performance comparison (regret mean with std. deviation) for resource provisioning.

**Performance analysis.** Table 2 summarizes the experimental results. In total, on all problem sizes (#equities), our method performs consistently and significantly better than both two-stage (L1 and L2) methods and the decision focused DF[9]. Among the three baselines, DF is significantly better than two-stage methods, similar to results in [3]. In fact, DF under this setting can be viewed as a variant of our method with infinite $K$ and no conversion of softening $\sum_i x_i = 1$. The comparison to DF also demonstrates the advantage of our method on processing such non-differentiable cases against the simple 2nd-order Taylor expansion. Besides, with the increment of the number of equities, regrets of all methods decease, which indicates that for the constraint $\sum_i x_i = 1$, larger number of equities brings smaller entries of $x$ on average (with the presence of $Q$, there are many non-zero entries of $x$), lowering the impact of prediction error for any single entry.

### 4.3 Resource Provisioning

**Problem setup.** We use ERCOT energy dataset [25], which contains hourly data of energy output from 2013 to 2018, 52535 data points in total. We use the last $20\%$ samples for test. We aim to predict the matrix $C \in \mathbb{R}^{24 \times 8}$, the loads of 24 hours in 8 regions. The decision vairable $x$ is 8-dimensional, and $d = 0.5 \times \mathbf{1} + 0.1N(0, 1)$. We test five sets of $(\alpha_1, \alpha_2)$, with $\alpha_1/\alpha_2$ ranging from 100 to 0.01.

**Experimental setup.** We use AdaGrad with learning rate of $0.01$, and clip the gradient with norm $0.01$. For the prediction model, we use McElWee's network [26] which was highly optimized for this task, with $(8 \times 24 \times 77)$-dimensional numerical features and embedding ones as input.

**Performance analysis.** Table 3 shows the experimental results. The absolute value of regret differs largely across different ratios of $\alpha_1/\alpha_2$. Our method is better than other methods, except for $\alpha_1/\alpha_2 = 1$, where the desired objective is exactly $L1$-loss and thus $L1$ performs the best. Interestingly, compared to L1/L2, the Weighted L1 performs better when $\alpha_1/\alpha_2 > 1$, but relatively worse otherwise. This is probably due to the dataset's inherent sample bias (*e.g.,* asymmetric distribution and occasional peaks), which causes the systematic bias (usually underestimation) of the prediction model. This bias exacerbates, when superposed with weighted penalty multipliers which encourage the existing bias to fall on the wrong side. Besides, the large variance for weighted L1 at $\alpha_1/\alpha_2 = 0.01$ is caused by a pair of outliers.

## 5 Related Work

**Differentiating `argmin`/`argmax` through optimality conditions.** For convex optimization problems, the KKT conditions map coefficients to the set of solutions, and thus can be differentiated for `argmin` using implicit function theorem. Following this idea, existing work developed implicit layers of argmin in neural network, including OptNet [13] for quadratic programs (QP) problems and CVXPY [14] for more general convex optimization problems. Further with linear relaxation and

QP regularization, Wilder *et al.* derived an end-to-end framework for combinatorial programs [9], which accelerates the computation by leverage the low-rank properties of decision vectors [3], and is further extended to mixed integer linear programs in MIPaaL [27]. Besides, for the relaxed LP problems, instead of differentiating KKT conditions, IntOpt [28] proposes an interior point based approach which computes gradients by differentiating homogeneous self-dual formulation.

**Optimizing surrogate loss functions.** Elmachtoub and Grigas [10] proposed a convex surrogate loss function, namely SPO+, measuring the decision error induced by a prediction, which can handle polyhedral, convex and mixed integer programs with linear objectives. TOPNet [29] proposes a learned surrogate approach for exotic forms of decision loss functions, which however is hard to generalize to handle constrained programs.

Differentiating argmin is critical for gradient methods to optimize decision-focused prediction models. Many kinds of efforts, including regularization for specific problems (e.g., differentiable dynamic programming [30], differentiable submodular optimization [31]), reparameterization [32], perturbation [33] and direct optimization ([12]), are spent for optimization with discrete or continuous variables and still actively investigated.

As a comparison, our work proposes a surrogate loss function for constrained linear and quadratic programs extended with soft constraints, where the soft constraints were not considered principally in previous work. Also, unlike OptNet [13] and CVXPY [15], our method does not need to solve KKT conditions. Instead, by adding reasonable costs for infeasibility, we convert the constrained optimization problem into an unconstrained one while keeping the same solution set, and then derive the required Jacobian analytically. To some degree, we reinvent the exact function [34] in prediction+optimization.

## 6   Conclusion

In this paper, we have proposed a novel surrogate objective framework for prediction+optimization problems on linear and semi-definite negative quadratic objectives with linear soft constraints. The framework gives the theoretical bounds on constraints' multipliers, and derives the closed-form solution as well as their gradients with respect to problem parameters required to predict by a model. We first convert the hard constraints into soft ones with reasonably large constant multipliers, making the problem unconstrained, and then optimize a piecewise surrogate locally. We apply and empirically validate our method on three applications against traditional two-stage methods and other end-to-end decision-focused methods. We believe our work is an important enhancement to the current prediction+optimization toolbox. There are two directions for the future work: one is to seek solutions which can deal with hard constraint parameters with negative entries, and the other is to develop the distributional prediction model rather than existing point estimation, to improve robustness of current prediction+optimization methods.

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
