# A Mathematical Proofs

Throughout the following proofs, for the convenience of description, we denote two symbols:

- the symbol $\angle(a, b)$, which means the angle of two vectors $a, b \in \mathbb{R}^n$;

- $cone(A)$, which means the conic combination of the row vectors of matrix $A$; $cone(A, B)$ stands for the conic combination of the union of the row vectors of matrix $A, B$.

Throughout this section, we will assume the same variables as in Eq.1-5 and Section 3 by default, and further make the following assumptions or simplification for ease of description.

- Assume the main objective $g(x, \theta) = \theta^T x$, where $||\theta||_2 \leq E$. In reality, elements of the predictive vector $\theta$ have a known range and thus $l_2$-norm of $\theta$ is bounded by a constant, namely $E \in \mathbb{R}$.

- We omit the soft constraints in original objective. Since all proofs in this section consider the worst direction of $\frac{\partial L}{\partial x}$ in Eq.5, for soft constraints which are linear, their derivatives are constant and can be integrated into $E$ for any bounded penalty multiplier $\alpha$.

- Similarly, we omit quadratic term $Q$. At first glance, it seems that $Q$ will bring unbounded derivative of decision variable $x$. However, note that we are finding the maximum value of function, and the matrix $Q$ is semi-definite positive, we have $\frac{\partial - x^T Q x}{\partial x_i} < 0$ for any $i$. Thus, denote the maximum point of $-x^T Q x$ to be $x_0$, we can find a finite radius $r > 0$, such that any point $x \in \mathcal{R}^n$ out of the circle with $x_0$ as the center and $r$ as radius is impossible to be the global optimal point; intuitively, the norm of the derivative of $Q$ becomes so large as to "draw the optimal point back" from too far, and thus we can ignore the points outside the circle. Since the circle is a bounded set, the norm of derivative $E$ is also bounded inside the circle.

- When we consider the effect of violating constraints, we only consider non-degraded activated cone, *i.e.,*, for an activated cone in a $n$-dimensional Euclidean space, there are at least $n$ non-redundant active constraints. Otherwise, as we stated in the main paper, we can project the activated cone $\mathcal{K} = \{z \in \mathbb{R}^n | z - x_0 \in cone(A')\}$ onto the complementary space of the cone tip $x_0$ (*i.e.,* the solution space of $A'$) and proceed on a low-dimensional subspace.

- As for constraint itself, we assume that for any row vector of $A$ in $Ax \leq b$, and any row vector of $B$ in $Bx = c$, we have $||A_i||_2 = 1$ and $||B_i||_2 = 1$ for any row vector $A_i, B_i$, $i \in \{1, 2, ..., m\}$ where $m$ is the number of constraints. If not, we can first normalize the row vectors of $A$ and $B$ while adequately scaling $b, c$, then apply our proofs.

Thus it is straightforward to apply our proofs to the general form of Eq.5.

## A.1 Lemma 1

**Lemma 1.** *(Bounding the utility gain) Let $R = f(x, \theta) - \max_{x_1 \in \mathcal{C}} f(x_1, \theta)$ be the utility gain, then $R \leq f(x, \theta) - f(x_0, \theta) \leq \frac{E}{\cos p_0} \sum_{i=1}^n (A_i' x - b_i')$, where $x$ is an infeasible point, $A'x \leq b'$ the active constraints at $x$, $p_0 = \angle(A_i'^*, \theta^*)$ where $A_i'^*$ and $\theta^*$ are the optimal solution of $\max_\theta \min_{A_i'} \cos\angle(A_i', \theta)$ (i.e., the maximin angle between $\theta$ and hyperplanes of the activated cone $\mathcal{K} = \{z \in \mathbb{R}^n | z - x_0 \in cone(A')\}$), and $x_0$ the projection of $x$ to the tip of cone $\mathcal{C} = \{z \in \mathbb{R}^n | A'z \leq b'\}$. $E$ is the upper bound of $||\theta||_2$.*

*Proof.* The first inequality is trivial as $x_0 \in \mathcal{C}$ and directly followed from the definition of utility gain $R$.

For the second inequality, by the law of sines, we have $||x - x_0||_2 = \frac{d_i}{\sin \tau_i}$, where $d_i$ is the distance of $x$ to the $i$-th hyper-plane in the activated cone $cone(A')$ with $A_i'$ (*i.e.,* the $i$-th row of $A'$) being the normal vector of the $i$-th hyper-plane in $A'x \leq b'$, and $\tau_i$ is the angle between $i$-th hyper-plane and $\theta$.

Then the utility gain $R$ from violating the constraints follows:

$$
\begin{aligned}
R &\le \theta^T(x - x_0) \\
&\le ||\theta||_2 ||x - x_0||_2 \\
&= ||\theta||_2 \frac{d_i}{\sin \tau_i} \quad \text{(By the law of sines; } i \text{ is arbitrary)} \\
&= ||\theta||_2 \frac{d_j}{\sin \tau_j} \quad \text{(By selecting } j = \operatorname*{argmin}_i \sin \tau_i = \operatorname*{argmax}_i cos\angle(A_i', x - x_0)) \quad (10) \\
&\le E \frac{\sum_{i=1}^n d_i}{\sin \tau_j} \quad \text{(E is the upper bound of } ||\theta||_2) \\
&= E \frac{\sum_{i=1}^n (A_i' x - b_i')}{\sin \tau_j} (d_i = \frac{|A_i' x - b_i'|}{||A_i'||_2} = A_i' x - b_i'; \text{ note the fifth assumption.})
\end{aligned}
$$

Thus, setting all entries of penalty multiplier $\beta$ in the hard constraint conversion to $O(\frac{E}{\sin \tau_j}) = O(\frac{E}{cos\angle(A_j', x - x_0)})$ will give us an upper bound of the minimum feasible $\beta$. $\qquad\square$

The rest of the work is to find the lower bound for $\min_\theta \max_i cos\angle(A_i', x - x_0)$ for any given activated cone $cone(A')$ and $x_0$; as we have assumed $||A_i'|| = 1 \ \forall i \in \{1, 2, ..., n\}$, the objective $\min_\theta \max_i cos\angle(A_i', x - x_0)$ can be equivalently written as[5]

$$
F = \min_\theta \max_i A_i' \theta \qquad (11)
$$

Deriving the lower bound of $F$ is the core part of proving Theorem 3 and 6.

## A.2 Lemma 7

In order to prove Theorem 2, we give a crucial lemma.

**Lemma 7.** *Consider a set of hyper-planes with normal vectors $A_1', A_2', ..., A_n' \in \mathbb{R}^n \ge 0$, $||A_i'||_2 = 1$ for $i \in \{1, 2, ..., n\}$ which forms a cone. Let $d_i$ be the distance of a point $x$ to the hyper-plane $A_i'$, where $x$ is in the cone of $A_i'$ (i.e., $x = k_1 A_1' + ... + k_n A_n'$, $k_i \ge 0 \ \forall i \in \{1, 2, ..., n\}$), then we have $||x||_2 \le \sum_{i=1}^n d_i$.*

*Proof.* Without loss of generality, let $x = k_1 A_1' + k_2 A_2' + ... + k_n A_n'$, and $||x||_2 = 1$ (we can scale $x$ if $||x||_2 \ne 1$). The distance $d_i = \frac{A_i'^T x}{||A_i'||_2} = A_i'^T x$. Therefore, we have

$$
\begin{aligned}
\sum_{i=1}^n d_i &= \sum_{i=1}^n A_i'^T(k_1 A_1' + ... + k_n A_n') \\
&= \sum_{j=1}^n k_j(A_j'^T A_j' + \sum_{i=1,i\ne j}^n A_i'^T A_j') \qquad (12) \\
&\ge \sum_{j=1}^n k_j ||A_j'||_2 \ (A_i' \ge 0, \text{ therefore } A_i'^T A_j' \ge 0) \\
&= k_1 + k_2 + ... + k_n
\end{aligned}
$$

We next prove that $k_1 + k_2 + ... + k_n \ge 1$. As $||x||_2 = 1$, we have

$$
\begin{aligned}
||x||_2 &= 1 \\
k_1 A_1'^T x + k_2 A_2'^T x + ... + k_n A_n'^T x &= 1
\end{aligned}
\qquad (13)
$$

As $A_i'^T x \in [0, 1]$, $k_1 + ... + k_n \ge 1$ must hold. Therefore, $\sum_{i=1}^n d_i \ge k_1 + ... + k_n \ge 1 = ||x||_2$. $\quad\square$

---

[5]With an abuse of notation, the $\theta$ in Eq.11 represents $x - x_0$, as the worst situation for deriving upper bound of $\beta$ is that $x - x_0$ has the same direction with the objective $\theta^T x$'s derivative $\theta$.

## A.3 Theorem 2

**Theorem 2.** *Assume the optimization objective $\theta^T x$ with constraints $Ax \le b$, where $A \ge 0$, and $b \ge 0$. Then, the utility gain $R$ obtained from violating $Ax - b \le 0$ has an upper bound of $O(\sum_i \max(w_i, 0)E)$, where $w = A'x - b'$, and $A'x \le b'$ is the active constraints.*

*Proof.* Let $d_i = (d_{i,1}, d_{i,2}, ..., d_{i,n})^T$ be the vector of distances to the $i$-th hyper-plane of $A'$ with normal vector $A'_i$ from $x$. By definition of $x$ and $x_0$, the distance vector of the point to the convex hull $\mathcal{C}$ is $x - x_0$. When $A \ge 0$ and $b \ge 0$, obviously Lemma 7 holds. Consider the active constraints $w = A'x - b' \ge 0$ which is the subset of active constraints in $Ax \le b$. The utility gain $R$ from violating the rules $w = A'x - b'$ satisfies:

$$
\begin{aligned}
R &\le \theta^T(x - x_0) \\
&\le ||\theta||_2 ||(x - x_0)||_2 \\
&\le ||\theta||_2 \sum_i ||d_i||_2 \text{ (Lemma 7)} \\
&\le E \sum_i ||d_i||_2 \\
&= E \sum_i \max(w_i, 0) \ (d_i = \frac{|A'_i x - b'_i|}{||A'_i||_2} = A'_i x - b'_i; \text{ note the fifth assumption.})
\end{aligned}
\tag{14}
$$

This holds for any $x$. Therefore, we turn the hard constraint $Ax \le b$ into a soft constraint with penalty multiplier $\beta = E$. □

## A.4 Theorem 3

**Theorem 3.** *Assume at least one constraint in $x \ge 0$ is active, then the utility gain $R$ by deviating from the feasible region is bounded by $O(\frac{n^{1.5} E \sum_i max(w_i, 0)}{\sin p_0})$, where $p_0 = \min_{i,j} \angle(A_i, e_j)$ (i.e., the smallest corner between axes and other constraint hyper-planes), and $w = \begin{bmatrix} A' \\ -I' \end{bmatrix} x - \begin{bmatrix} b' \\ 0 \end{bmatrix}$, where $A'x \le b'$ and $-I'x \le 0$ are active constraints in $Ax \le b$ and $x \ge 0$ respectively.*

*Proof.* As Lemma 1 implies, the key point of proving this theorem is to prove that $\theta$ (the derivative of the objective), is always close to some rows of $A'$ and $-I'$.

Consider the activated cone $\mathcal{K} = \{z \in \mathbb{R}^n | z - x_0 \in cone(A', -I')\}$; all hyper-planes of $A'x = b'$ and violated entries of $x \ge 0$ must pass through $\mathcal{K}$'s tip $x_0$. Figure 3 gives such an illustrative example. For the rest of the proof, it is enough to assume that the activated cone $\mathcal{K}$ is a non-degraded cone, since otherwise, as the main paper states, we can project the activated cone onto the supplementary space of its tip. By the projection, we actually reduce the problem to the same one with lower dimensions of $x$, and can apply the following proof in the same way. Let $q$ be the number of rows in $I'$, i.e., the number of active constraints in $x \ge 0$; and $r$ be the number of rows in $A'$, i.e., the number of active constraints in $Ax \le b$. Since $\mathcal{K}$ is non-degraded, we have $r + q \ge n$.

Without loss of generality, we transform the problem equivalently by rearranging the order of dimensions of $x$ so that $I'$ is the first $q$ rows of $I_{n \times n}$. Let's see an example after this transformation. As shown in Figure 3, the three solid black vectors are $(-1, 0, 0)$, $(0, -1, 0)$, and $A'_1 \ge 0$, where the first and second entries of $A'_1$ are strictly greater than 0 (otherwise the cone is degraded). Similarly, for the $n$-dimensional non-degraded activated cone, we consider the maximin angle between $\theta$ and all active inequality constraints $A'$ and $-I'$ (which is equivalent to finding the minimax cosine value of angles between $\theta$ and all active inequality constraints). Then, guided by Equation 11 in Lemma 1, the upper bound of distance between $\theta$ and hyper-planes with normal vector $A'_j$, is the solution of the following optimization problem w.r.t. $\theta \in \mathbb{R}^n$ and $A'_j$ (since the cone is non-degraded, we assume that $A'_j$ are linearly independent):

$$
F = \min_{\theta} \max_{j \in J = \{1, ..., r\}, k \in K = \{1, ..., q\}} \{A'_j \theta, -e_k^T \theta\},
$$
$$
s.t. \ ||\theta||_2 = 1, \theta \in \mathcal{K}, r + q \ge n.
\tag{15}
$$

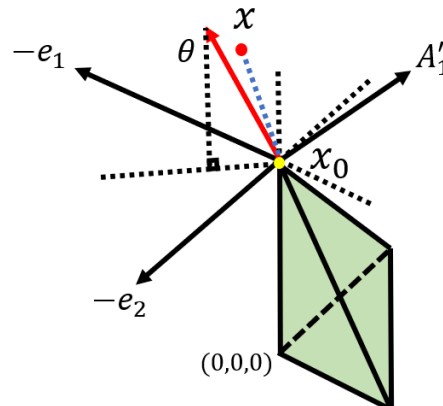

Figure 3: A 3-dimensional illustration of the proof for Theorem 3. The green polytope is the feasible region; the black solid vectors are the related vectors (i.e. rows of $A'$) and the red line is the utility vector $\theta$. Apparently, $(0,0,1)^T\theta \geq 0$, otherwise the intersection point will move down to origin. The worse case appears when the red $\theta$ is on the circumcenter of the triangle formed by the three solid unit vectors.

where $e_k$ is the unit vector where the $k$-th entry is 1 and others are 0. Note that according to our presumptions, $A'_j \geq 0$, $||A'_j||_2 = 1$ for $j \in \{1, 2, ..., r\}$ have already satisfied. The constraints $\theta \in \mathcal{K}$ come from the following consideration: if with given $\theta$ we update $x$ in an iterative manner such as gradient ascending, $x$ would eventually leave current activated cone where $\theta_i < 0$ for $i \in \{q+1, ..., n\}$ (see Figure 3 for an illustration). Since for any dimension index $i \in \{q+1, ..., n\}$, all entries of $A'_j$ and $-e_k$ ($j \in J, k \in K$) are non-negative, we know that $\theta_i \geq 0$, $\forall i \in \{q+1, ..., n\}$. To further relax the objective $F$ and derive the lower bound, we will assume $r + q = n$ in the rest of the paper; if $r \geq n - q$, we can simply ignore all $A'_j$ in the max operator with $j > n - q$.

Given the signs of $\theta$, we next derive a lower bound of $F$ with respect to any given $A'_j$ and set of $k$. Now we relax $F$ by setting all entries of $A'_j$ to 0, except for the entries with index $\{1, 2, ..., q\}$ and $s_j$ to get $A''_j$, where $\{s_j\}(j \in \{q+1, ..., n\})$ is a permutation of $\{q+1, q+2..., n\}$. We have:

$$F \geq \min_{\theta} \max_{j \in \{1,..,r\}, k \in \{1,..,q\}} \{A''^T_j \theta, -e^T_k \theta\} \text{ (Relaxing within the max operator; note the sign of } \theta)$$

$$\geq \min_{\theta} \max_{j \in \{1,..,r\}, k \in \{1,..,q\}} \{\sum_{i=1}^{q} A'_{j,i}\theta_i + A'_{j,s_j}\theta_{s_j}, -\theta_k\}$$

$$\geq \min_{\theta} \max_{s_j \in \{q+1,n\}, k \in \{1,..,q\}} \{\sum_{i=1}^{q} \theta_i + \alpha_0\theta_{s_j}, -\theta_k\}$$

(16)

where $||\theta||_2 = 1$, $\alpha_0$ is the smallest non-zero entry among $A'_{j,s_j}$ (and note that $A \geq 0$ and $\theta_{s_j} \geq 0$, which means $\alpha_0\theta_{s_j} \geq 0$). The inequalities hold for the relaxation within the max operator. Note that the last line of inequality assumes $\theta^*_i \leq 0$ for $i \in \{1, 2, ..., q\}$ for the optimal $\theta^*$; otherwise, we can scale further, ignore the index $i$ of the optimal point $\theta^*$ where $\theta^*_i > 0$ and proceed with $n-1$ dimensions for the following two facts:

1. $\sum_{i=1}^{q} \theta_i + \alpha_0\theta_{s_j}$ term becomes smaller after ignoring such dimension;

2. $\theta \in \mathcal{K}$, which means for any $\theta$ and any vector $y$ of the cone we have $\theta^T y \geq 0$, and the last line still corresponds to a cone. Therefore the optimal value of last line is no less than 0; the removal of $-\theta_k$ term given $\theta_k > 0$ will not affect the optimal value.

Then, according to the property of minimax, $\sum_{i=1}^{q} \theta_i + \alpha_0\theta_{s_j}$ should be equal to $-\theta_k$; Otherwise, if $\sum_{i=1}^{q} \theta_i + \alpha_0\theta_{s_j}$ is larger than $-\theta_k$, we can adjust the value of $\sum_{i=1}^{q} |\theta_i|$ and $\sum_{j=q+1}^{n} |\theta_j| = \sum_{j=q+1}^{n} \theta_j$ by a small amount such that the result is better and $||\theta||_2 = 1$ is still satisfied, which can

be repeated until no optimization is possible, and vice versa. With $\sum_{i=1}^{q} |\theta_i|$ and $\sum_{j=q+1}^{n} \theta_j$ fixed, the entries of $\theta$ are equal to each other within each group by symmetry.

Therefore, the solution $\theta$ should be in the form of $(\frac{c_1}{\sqrt{n}}, \frac{c_1}{\sqrt{n}}, ..., \frac{c_2}{\sqrt{n}}, \frac{c_2}{\sqrt{n}})$ where $c_1 < 0, c_2 > 0$, and we have the following set of equations:

$$qc_1^2 + (n-q)c_2^2 = n$$
$$qc_1 + \alpha_0 c_2 = -c_1 \tag{17}$$

With equations in Eq. 17, we get $c_2 = -(1 + \alpha_1 q)c_1/\alpha_0$, $c_1^2(q + (n-q)\frac{(1+q)^2}{\alpha_0^2}) = n$. As $q \leq n$, and as $||\alpha_j|| = 1$, $\alpha_0$ can be further relaxed to the sine of the smallest angle $p_0$ between axes $x \geq 0$ and the other inequality constraints, and thus we have the lower bound $F \geq \frac{c_1}{\sqrt{n}} = O(\sin p/n^{1.5})$ for the optimization problem listed in Equation 16, which is also the cosine lower bound to the nearest normal vector $A'_j$ and $-e_k$ of the activated cone. This is the denominator of the desired result; the final result follows as we apply such bound to Lemma 1. $\square$

### A.5 Corollary 4

**Corollary 4.** *For binary constraints where the entries of $A$ (before normalization) are either $0$ or $1$, the utility gain $R$ of violating $x \geq 0$ constraint is bounded by $O(n^{1.5}E\sum_i \max(w_i, 0))$, where $w$ is the same as Theorem 3.*

*Proof.* The proof of Corollary 4 is almost the same with Theorem 3, except that if the constraint $A$ is cardinal, then each entry of $A$ is either $0$ or $1$. Thus, as all the non-zero entry are the same, we shall replace the third line in Equation 16 with

$$\min_{\theta} \max_{j} \{\frac{\sum_{i=1}^{q} \theta_i}{\sqrt{h_j}} + \frac{\theta_{s_j}}{\sqrt{h_j}}, -\theta_k\} \tag{18}$$

where $h_j$ is the number of non-zero entry of $A'_j$, each entry being $\frac{1}{\sqrt{h_j}}$ as normalized $||A'_j||_2 = 1$.

By substituting $h_j$ with $n^6$, this optimization problem can be further relaxed to

$$\min_{\theta} \max_{j} \{\frac{\sum_{i=1}^{q} \theta_i}{\sqrt{n}} + \frac{\theta_{s_j}}{\sqrt{n}}, -\theta_k\} \tag{19}$$

and we can change Equation 17 to

$$qc_1^2 + (n-q)c_2^2 = n$$
$$qc_1 + c_2 = -\sqrt{n}c_1 \tag{20}$$

Therefore, similar to the proof of Theorem 3, we get $\frac{c_1}{\sqrt{n}} = \sqrt{\frac{1}{q+(n-q)(\sqrt{n}+q)^2}}$, which leads to the bound $O(n^{1.5}E\sum_i \max(w_i, 0))$, removing the $\sin p$ in the denominator. $\square$

### A.6 Theorem 5

**Theorem 5.** *If there is only one equality constraint $B^T x = c$ (e.g. $\sum_i x_i = 1$) and special inequality constraints $x \geq 0$, $Ix \leq b$, then the utility gain $R$ from violating constraints is bounded by $O(\frac{n^{1.5}E\sum_i \max(w_i, 0)}{\sin p})$, where $p$ is the same with theorem 3, $w$ is the union of active $B^T x - c$ and $-x$.[7]*

*Proof.* We first prove the situation where the inequality constraint is only $x \geq 0$. If we only have one equality constraint, then it can be seen as two separate inequality constraints, which are $B^T x \leq c$ and $-B^T x \leq -c$; for any non-degraded activated cone, as $B^T x < c$ and $B^T x > c$ cannot hold simultaneously, there is at most one normal vector of constraint in the activated cone.

---

[6]Note in Theorem 3 we have already mentioned the non-positivity of $\theta_i$ where $i \in \{1, 2, ..., q\}$, thus bigger $h_j$ brings smaller objective.

[7]See Theorem 3 for the meaning of union.

If $B^T x \leq c$ is active (*i.e.,* this constraint is violated, now we have $B^T x > c$), then with $B \geq 0$, the case is exactly the same with Theorem 3. Otherwise, if $-B^T x \leq -c$ (*i.e.,* $B^T x \geq c$) is active (*i.e.,* $B^T x \geq c$ is violated, now $B^T x < c$), then the normal vector of the current active constraint $-B^T x \leq -c$ is $-B \leq 0$. Note that all other normal vectors of active constraints are $-e_k \leq 0$ (which is the same situation as that in Equation 15 in the proof of Theorem 3, except that $A'_j$ is substituted with $-B^T \leq 0$); this indicates that the condition of Lemma 7 is satisfied under such scenario. Therefore, we can apply the proof of Theorem 2 in this case and get a better bound than that in the first scenario.

Then, the full theorem is proved as follows: consider any activated cone $\mathcal{K}$. For any $i \in \{1, 2, ..., n\}$, $x_i \leq b_i$ and $x_i \geq 0$ cannot be active simultaneously. If the former is activated, we replace $x_i$ with $b_i - x_i$; if the latter (or neither) is active, we remain $x_i$ as normal. Then, for this activated cone, the scenario is exactly the same with the situation where the inequality constraints are only $x \geq 0$. Similarly, if the equality constraint is cardinal, with the same proof of Corollary 4, we can remove the $\sin p$ in the denominator.

$\square$

## A.7 Theorem 6

**Theorem 6.** *Given constraints $Ax \leq b$, $x \geq 0$, and $Bx = c$, where $A, B, b, c \geq 0$, the utility gain $R$ obtained from violating constraints is bounded by $O(\sqrt{n}\lambda_{max} \sum_i \max(w_i, 0))$, where $\lambda_{max}$ is the upper bound for eigenvalues of $P^T P$ ($\mathcal{P} : x \to Px$ is an orthogonal transformation for an $n$-sized subset of normalized row vectors in $A, B$ and $-I$), and $w$ is the union of all active constraints from $Ax \leq b$, $x \geq 0$, $Bx \leq c$ and $-Bx \leq -c$.*

*Proof.* For any non-degraded activated cone $\mathcal{K}$, we have the following optimization problem (it is worth noting that reformulating our original problem to this optimization one is inspired by Equation 11 in Lemma 1):

$$
\begin{aligned}
F = \min_\theta \max_{j \in J, k \in K} \{ \max_{i \in D, n_i \in \mathbb{R}^n} \{n_i^T \theta\}, A'_j \theta, -e_k^T \theta \} \\
s.t. ||\theta||_2 = 1, \theta \in \mathcal{K} \\
d + r + q \geq n, ||n_i||_2 = 1 \ \forall i \in D = \{1, 2, ..., d\}, < B_1, ..., B_d > = < n_1, ..., n_d >
\end{aligned}
\tag{21}
$$

where $A'_j$ is the normal vector of the $j$-th inequality constraints in $Ax \leq b$, $n_i$ is the $i$-th normal vector for the subspace of $Bx = c$ (and thus represents the same subspace that is represented by the rows of $B$). According to the assumption, we scale $Ax \leq b$ and $Bx = c$ so that $||n_i|| = ||A'_j|| = 1$ for any $i, j$. Note that different from the optimization problem stated in Theorem 3, in this problem $q$ can be 0, which means that there can be no entry of $x \geq 0$ active. To get the lower bound of $F$, we may relax the function by keeping the normal vectors $\{n_i\}$ in their original direction $B$ to $F \geq G = \min_\theta \max_{i \in D, j \in J, k \in K} \{B'_i \theta, A'_j \theta, -e_k^T \theta\}$ where the $i$-th row of $B$ satisfies $B'_i = B_i$ or $B'_i = -B_i$ (the sign of $B'_i$ is decided by the direction of $\theta$), and thus $B'_i \geq 0$ or $B'_i \leq 0$ (*i.e.,* any pair of elements in $B'_i$ will not have opposite signs) for any $i$. Moreover, $||B'_i||_2 = 1$. We denote the positive $B'_i$ as $B^p_{i_1}$, $i_1 \in D_1$ and the negative $B'_i$ as $B^n_{i_2}$, $i_2 \in D_2$. Then similar to Theorem 3,

$$
G = \min_\theta \max_{i_1 \in D_1, i_2 \in D_2, j \in J, k \in K} \{B^n_{i_1} \theta, B^p_{i_2} \theta, A'_j \theta, -e_k^T \theta\}
\tag{22}
$$

where $D_1 \bigcap D_2 = \emptyset$, $D_1 \bigcup D_2 = D$. we aggregate $A'$, $B^p$ and $-e_k$, $B^n$ into $\alpha$, and get

$$
G = \min_\theta \max_{j \in S} \{\alpha_j^T \theta\}
\tag{23}
$$

where $|S| \geq n$, $\alpha_j \geq 0$ or $\leq 0$ for any $j \in S$. Therefore, we can further relax $G$ by selecting $n$ linearly independent vectors that are the closest to the minimum product[8] and ignore the others for the $\max$ operator, and for the rest of the proof we may assume that $|S| = n$.

---

[8]The existence of such set of vectors comes from the non-degradation of the activated cone.

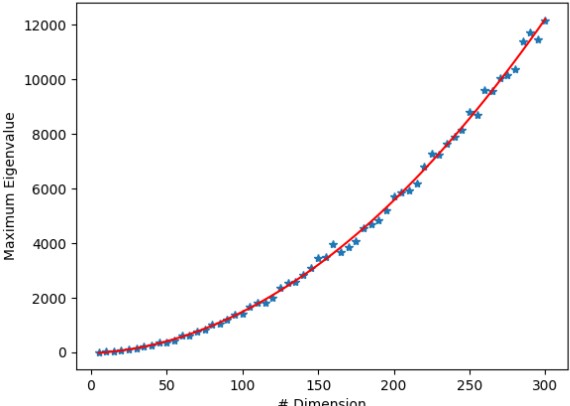

Figure 4: Empirical estimation of the max maximum eigenvalue over 1000 trials with $D$ being uniform. The $x$ axis is $n$, and the $y$ axis is $\lambda_{max}$. The 2-degree polynomial fitting curve is $0.1282x^2 + 2.392x - 32.89$.

We will next consider a linear transformation $\mathcal{P}: x \to Px$ from $\mathbb{R}^n$ to $\mathbb{R}^n$ that transforms $\{\alpha_j\}$ into an orthogonal normal basis. Then we have

$$
\begin{aligned}
\lambda_{max} G &= \min_\theta \max_j \lambda_{max} \alpha_j^T \theta \\
&\geq \min_\theta \max_i \alpha_j P^T P \theta \\
&= \frac{1}{\sqrt{n}} \sum_{k=1}^n \alpha_j P^T P \alpha_k \\
&= \frac{1}{\sqrt{n}} \alpha_j P^T P \alpha_j \\
&= \frac{1}{\sqrt{n}}
\end{aligned}
\tag{24}
$$

where $\lambda_{max}$ is the maximum eigenvalue of $P^T P$. The row vectors of $A''$ are $\{\alpha_i'\}$. Therefore, our problem $G$ has a lower bound of $\frac{1}{\sqrt{n}\lambda_{max}}$ where $\lambda_{max}$ is the maximum eigenvalue for $P^T P$; $A''$ consists of $n$ of the vectors in $A'$ and has the largest maximum eigenvalue for $P^T P$. $\qquad \square$

Though we do not derive the bound of $\lambda_{max}$ with respect to the number of dimension $n$ and the angle $p_0$ between hyper-planes in the activation cone, we empirically evaluate the behavior of $\lambda_{max}$ with respect to $n$ on randomly generated data. We first generate a normal vector $n \in R^n$, with each entry generated independently at random from the distribution $D$; then we generate $n$ vectors $\{\alpha_1, ..., \alpha_n\}$ in $R^n$ with their entries either all not smaller than 0 or all not greater than 0; the orthant is chosen with probability $0.5$. Each entry is independently generated from $D$ and shifted by a constant to enforce the signs. We ensure that $\forall i, \alpha_i^T n \geq 0$ by discarding the vectors that do not satisfy such constraint. $D$ can be uniform distribution $U(0, 1)$, the distribution for absolute value of normal distribution $|N(0, 1)|$, or beta distribution $B(2, 2)$. The $\alpha_i$ is then normalized, and we record the largest eigenvalue of $R^T R$, where $A = \begin{bmatrix} \alpha_1^T \\ \alpha_2^T \\ ... \\ \alpha_n^T \end{bmatrix}$, $A = RQ$ is the RQ decomposition of $A$. We repeat 1000 times for each $n$ and record the mean and maximum eigenvalue for $R^T R$. Below is the result of our evaluation; it shows that the maximum eigenvalue $\lambda_{max}$ is approximately $O(n^2)$. See Figure 4,5, and 6 for illustration.

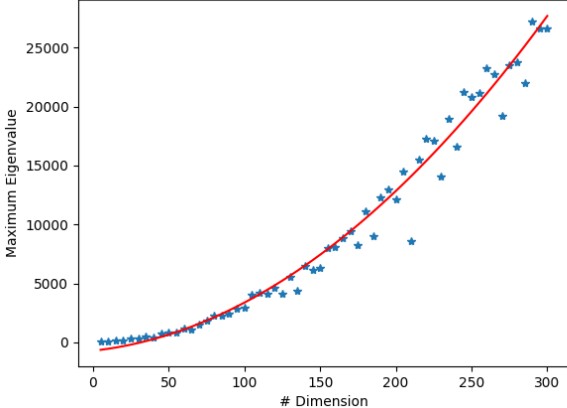

Figure 5: Empirical estimation of the max maximum eigenvalue over $1000$ trials with $D$ being Gaussian. The $x$ axis is $n$, and the $y$ axis is $\lambda_{max}$. The 2-degree polynomial fitting curve is $0.2693x^2 + 13.95x - 716.8$.

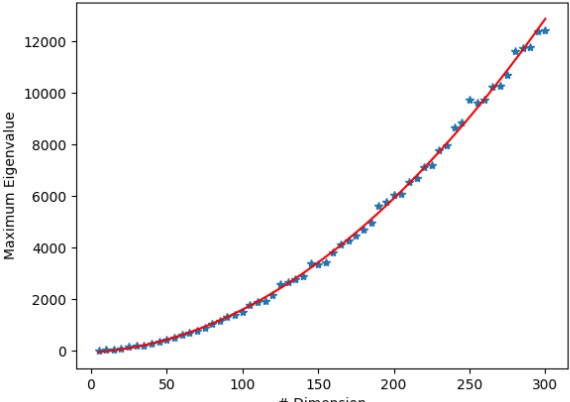

Figure 6: The max maximum eigenvalue over $1000$ trials with $D$ being $Beta(2,2)$. The $x$ axis is $n$, and the $y$ axis is $\lambda_{max}$. The 2-degree polynomial fitting curve is $0.1325x^2 + 3.403x - 69.95$.

## B    Choices of Surrogate $\max$ Functions

The final goal of surrogate $max$ function is to relax the objective with the term $\alpha^T \max(z = Cx - d, 0)$, making it differentiable over $\mathbb{R}^n$. Such objective is a piecewise function with respect to $z$ with two segments: one is constant $0$ with derivative $0$, the other is linear with a constant derivative.

At first glance, sigmoidal surrogates are seemingly the most straightforward candidate for modeling the derivative of such a piecewise function. For a sigmoidal approximation of soft constraints, $S(z)$ should satisfy the following four conditions, among which the first two are compulsory, and the third and fourth can be slightly altered (e.g. by setting $\epsilon_2 = 0$ and remove the fourth condition).

1. When $z \to \infty$, $S'(z) \to M^- = (1 + \epsilon_1)^-$. $\epsilon_1 > 0$ should be a small amount, and it serves as a perturbation since otherwise $g'(z)$ would be always greater than $0$. However, $\epsilon_1$ should not be too small to let the optimal point be too far away from the original constraint (for the gradient will reach $\alpha$ too late).

2. $S$ must be differentiable, and must have a closed-form inverse function.

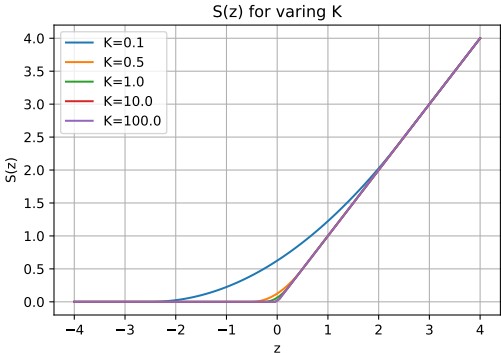

Figure 7: S(z), the proposed surrogate function of $max(z, 0)$

3. When $z \to -\infty$, $S'(z) \to \epsilon_2^+ < 0$, where $\epsilon_2$ is constant and very close to 0. The $S'(z)$ should cause as small impact as possible when $z < 0$ (when there is no waste), and meanwhile it should (very slightly) encourage $z$ to grow to the limit instead of discouraging.

4. When $z = 0$, $S'(z) = 0$, which means the penalty function $S$ has no influence on the clipping border. This condition is important for linear programming; it can be removed for quadratic programming.

To satisfy the conditions above, we need a closed-form sigmoidal function with closed-form inverse function to be $S'(z)$, the *derivative* of the surrogate. Such function can be Sigmoid function ($S'(z) = \frac{1}{1+e^{-z}}$), Tanh/arctan function, or fractional function ($S'(z) = 0.5(\frac{z}{\sqrt{z^2+1}} + 1)$), or equivalently $S(z) = 0.5\sqrt{z^2 + 1} + 0.5z$). With the form of $S(z)$ confirmed, our function should look like[9] $L(x) = g(x, \theta) - \alpha^T S(z)$, with the derivative $\frac{\partial L}{\partial x} = \frac{\partial g}{\partial x} - \frac{\partial z}{\partial x} S'(z)\alpha$.

Take $g(x) = \theta^T x$, $S(z) = 0.5\sqrt{1 + z^2} + 0.5z$ as an example. The optimal point satisfies

$$\theta = C^T \text{diag}(\frac{0.5z}{(1 + z^2)^{0.5}} + 0.5)\alpha \tag{25}$$

Let $y_i = (\frac{0.5z}{(1+z_i^2)^{0.5}} + 0.5)\alpha_i$, $w_i = (\frac{0.5x}{(1+x_i^2)^{0.5}} + 0.5)$, then $\theta = C^T y$. $S(z)$ can be substituted by any function with sigmoidal function (e.g. $arctan, tanh$, and $sigmoid$) as its derivative. However, $C$ is impossible to be invertible, for $C$ includes an identity matrix representing $x \geq 0$ and must have strictly more rows than columns if $C$ has any other constraint. Therefore, we cannot solve $z$ by first solving $y$ exactly in a linear system; we need to solve the equation of fractional function, or estimate $y$ with pseudo-inverse. Hence, the closed-form solution with sigmoidal surrogate is at least non-attractive.

To derive a closed-form solution $x$ for the equation $0 = \frac{\partial L}{\partial x} = \frac{\partial g}{\partial x} - \frac{\partial z}{\partial x} S'(z)\alpha$, $S'(z)$ should be in a simple form (e.g. polynomial form with degrees lower or equal to 4 with respect to $x$). Unfortunately, no basic function except the sigmoid function can satisfy the conditions listed above; the degree limit makes higher-order Taylor expansion infeasible. Thus, our only choice left is to find a differentiable piecewise function with linear parts on both sides; we shall first solve the optimal point of the surrogate numerically, and extend the segment where the optimal point locates to $R^n$; the gradient $\frac{\partial x}{\partial \theta}$ is correct due to the uniqueness of optimality (which requires a convex surrogate).

Therefore, we choose the function $S(z)$ to be

$$S(z) = \begin{cases} 0 & \text{if } z < -\frac{1}{4K} \\ K(z + \frac{1}{4K})^2 & \text{if } -\frac{1}{4K} \leq z \leq \frac{1}{4K} \\ z & \text{if } z \geq \frac{1}{4K} \end{cases} \tag{26}$$

---

[9] To satisfy the requirements above, the function needs scaling and translation; we omit them for the simplicity of formulae by simply set $\alpha = 1$, $\epsilon_1 = \epsilon_2 = 0$.

as illustrated in Figure 7, where $K$ is a constant. The surrogate is chosen for its simplicity. Not only does such choice satisfy our conditions with minimum matrix computation, but moreover, $S'(z)$ is linear, which means the equation can be solved in a linear system. The limitation of such method is that it introduces a hyper-parameter $K$ to tune: when $K$ is large, if the predicted optimal point is far from ground-truth, the gradient $\frac{\partial f(x, \theta_{\text{real}})}{\partial x}$ will be too large, making the training process instable. On the other hand, if $K$ is small, the optimal point may deviate from the feasible region too far, as the hard constraints are not fully enforced within the region $z \in [-1/4K, 4K]$. Fortunately, our empirical evaluation shows that the effect of $K$ is smooth, and can be tuned by a grid search.

## C    Learning and Inference Algorithm

In general, the learning and inference procedures based on our method have the same prediction+optimization workflow as the computational graph we gave in Fig.1. In this section, we propose the detailed algorithm with emphasis on the specific steps in our method.

Alg. 1 is the learning algorithm of stochastic gradient descent in mini-batch. For each sample $(\xi_i, \theta_i)$, it predicts the context variables as $\hat{\theta}_i$ (Statement 3) and then solve the program $f$ to get the optimal decision variable as $\hat{x}_i^*$ (Statement 4). Note that here $f$ is solved by solvers such as Gurobi and CPLEX that are capable to get the deterministic optimal solutions. With $\hat{x}_i^*$, the piece of $S(z)$ is determined, and thus the surrogate objective $\bar{f}$. From Statement 5 to 7, we decide the form of surrogate $\bar{f}$ for $\hat{x}_i^*$ by first calculating $\hat{z}_i^*$, then use $\hat{z}_i^*$ to decide $M_i$ and $U_i$ in the surrogate. $[\cdot]$ is an element-wise indicator function. Specially, if $C$ or $d$ is the predicted parameter, $\hat{z}_i^*$ should be calculated by the predicted value of $C$ or $d$; moreover, in statement 8 we should calculate $\frac{\partial r}{\partial x}$ on the estimated segment, but with ground-truth value of $C$ or $d$. In Statement 8, we compute the gradient by back-propagation. Specifically, we directly compute $\frac{\partial x}{\partial \theta}$ and $\frac{\partial r}{\partial x}$ by their analytical forms, while $\frac{\partial \theta}{\partial \psi}$ by auto-grad mechanism of end-to-end learning software such as PyTorch [17], which we use throughout our experiments. Finally, in Statement 9, we update the parameters of $\psi$ with accumulated gradients in this mini-batch.

---

**Algorithm 1:** SGD Learning with the surrogate $\bar{f}$

**Input**   : dataset $D = \{(\xi_i, \theta_i)\}_{i=1}^N$
**Input**   : optimization settings $\{A, b, C, d, \alpha\}$
**Input**   : derived hyper parameters $\{K\}$
**Input**   : learning rate $a$ and batch size $s$
**Output** : the predictive model $\Phi_\psi$ with parameters $\psi$
**begin**

1    Sample mini-batch $D_k \sim D$;
2    **foreach** $(\xi_i, \theta_i) \in D_k$ **do**
        // Estimate $\theta_i$ with the model $\Phi_\psi$.
3        $\hat{\theta}_i \leftarrow \Phi_\psi(\xi_i)$;
        // Solve the original $f$ with $\hat{\theta}_i$.
4        $\hat{x}_i^* \leftarrow \arg\max_x f(x, \hat{\theta}_i)$;
        // decide surrogate $\bar{f}$ for $\hat{x}_i^*$ by calculating $M_i$ and $U_i$
5        $\hat{z}_i^* \leftarrow C\hat{x}_i^* - d$;
6        $M_i \leftarrow diag([-1/4K \leq \hat{z}_i^* \leq 1/4K])$;
7        $U_i \leftarrow diag([\hat{z}_i^* \geq 1/4K])$;
        // Compute $\frac{\partial \bar{r_i^*}}{\partial \psi}$, where $\bar{r_i^*} = \bar{f}(\hat{x}_i^*, \theta_i)$
8        $\frac{\partial \bar{r_i^*}}{\partial \psi} \leftarrow \frac{\partial \hat{\theta}_i}{\partial \psi} \times \frac{\partial \hat{x}_i^*}{\partial \hat{\theta}_i} \times \frac{\partial \bar{r_i^*}}{\partial \hat{x}_i^*} |_{\theta_i, \hat{\theta}_i, \hat{x}_i^*}$
    // Update $\psi$ with ascent gradients
9    $\psi \leftarrow \psi + a \times \frac{1}{s} \times \sum_{i=1}^s \frac{\partial \bar{r_i^*}}{\partial \psi}$

---

Alg. 2 is the program with context variable prediction for inference; it is the same with the traditional predict-then-optimize paradigm. Note that the surrogate is no longer needed in this phase.

---

**Algorithm 2:** optimization with predictive variables

---

**Input** : dataset $D = \{\xi_i\}_{i=1}^N$
**Input** : optimization settings $\{A, b, C, d, \alpha\}$
**Input** : derived hyper parameters $\{K\}$
**Input** : learned predictive model $\Phi_\psi$
**Output** : Solutions $\{(\hat{x_i^*}, \hat{r_i^*})\}$
**begin**
1    **foreach** $\xi_i \in D$ **do**
      // Estimate $\theta_i$ with the model $\Phi_\psi$.
2       $\hat{\theta}_i \leftarrow \Phi_\psi(\xi_i)$;
      // Solve $f$ with the estimated $\hat{\theta}_i$.
3       $\hat{x_i^*} \leftarrow \arg\max_x f(x, \hat{\theta}_i)$;
      // Evaluate $f$ with the real $\theta_i$.
4       $\hat{r_i^*} \leftarrow f(\hat{x_i^*}, \theta_i)$;

---

## D   Derivation of Gradients

In this section, we will show the derivation process of the gradient with respect to the three problems stated in the main paper.

### D.1   Linear Programming with Soft Constraints

The problem formulation is:

$$\max_x \theta^T x - \alpha^T \max(Cx - d, 0), \text{ s.t. } Ax \le b \tag{27}$$

where $\alpha \in \mathbb{R}^n \ge 0, A \in \mathbb{R}^{m_1 \times n} \ge 0, b \in \mathbb{R}^{m_1} \ge 0$, and $\theta \in \mathbb{R}^n$ is to be predicted. In this formulation and the respective experiment (synthetic linear programming), we assume that there is no equality constraint.

Let $C' = \begin{bmatrix} C \\ A \\ -I \end{bmatrix}, d' = \begin{bmatrix} d \\ b \\ 0 \end{bmatrix}, \gamma = \begin{bmatrix} \alpha \\ O(\frac{n^{1.5}E}{\sin p}) \\ O(\frac{n^{1.5}E}{\sin p}) \end{bmatrix}$, where $E$ is the upper bound of $||\theta||_2$, $p$ is the minimum angle between hyper-planes of $Ax \le b$ and the axes. Then, we can write the surrogate as

$$\gamma^T S(z) = \gamma^T (0.5M(z + \frac{1}{4K})^2 + Uz) \tag{28}$$

Calculating the derivative of $\theta^T x - \gamma^T S(z)$ at the optimal point, we get

$$\theta = C'^T (M(\text{diag}(z) + \frac{1}{4K}I) + U)\gamma = C'^T M \text{diag}(\gamma)(C'x - d') + C'^T (\frac{1}{4K}M + U)\gamma \tag{29}$$

This equation gives us the analytical solution of $x$:

$$x = (C'^T M \text{diag}(\gamma)C')^{-1}(\theta + C'^T M \text{diag}(\gamma)d' - C'^T (\frac{1}{4K}M + U)\gamma) \tag{30}$$

Based on such derivative solution of $x$, we differentiate with respect to $\theta$ on both sides:

$$\frac{\partial x}{\partial \theta} = (C'^T M \text{diag}(\gamma)C')^{-1} \tag{31}$$

On the other hand, given real parameter $\theta_{real}$, we calculate the derivative of $f(x, \theta_{real})$:

$$\frac{\partial f(x, \theta_{\text{real}})}{\partial x} = \theta_{\text{real}} - C'^T M \text{diag}(\gamma)(C'x - d') - C'^T (\frac{1}{4K}M + U)\gamma \tag{32}$$

## D.2 Portfolio Selection with Soft Constraints

We consider minimum variance portfolio ([16]) which maximizes the return while minimizes risks of variance. The problem formulation is:

$$\max_x \theta^T x - x^T Q x - \alpha^T \max(Cx - d, 0)$$

$$\text{s.t. } x^T \mathbf{1} = 1, \ x \geq 0, \ Q \geq 0, \ \alpha \geq 0$$

(33)

where $x \in \mathbb{R}^n$ is the decision variable vector – equity weights, $\theta \in \mathbb{R}^n$ is the equity returns, the semi-definite positive $Q \in \mathbb{R}^{n \times n}$ is the covariance matrix of returns $\theta \in \mathbb{R}^n$. Equivalently, we rewrite the constraints of $x$ to fit our surrogate framework, as $Ax \leq b$ where $A = \begin{bmatrix} -\mathbf{1}_{1 \times n} \\ \mathbf{1}_{1 \times n} \\ -I \end{bmatrix}, b = \begin{bmatrix} -1 \\ 1 \\ 0 \end{bmatrix}$.

Let $C' = \begin{bmatrix} C \\ A \end{bmatrix}, d' = \begin{bmatrix} d \\ b \end{bmatrix}, \gamma = \begin{bmatrix} \alpha \\ O(n^{1.5}E)\mathbf{1} \end{bmatrix}$, and we have surrogate $\gamma^T S(z) = \gamma^T (0.5M(Z + \frac{1}{4K})^2 + Uz)$. Then, we may derive the optimal solution $x$ and the gradient of $f(x, \theta_{\text{real}}, Q_{\text{real}})$ with real data with respect to $x$ as

$$x = (2Q + C'^T M \text{diag}(\gamma) C')^{-1}(\theta + C'^T M \text{diag}(\gamma) d' - C'^T (U + \frac{M}{4K})\gamma)$$

$$\frac{\partial f(x, \theta_{\text{real}}, Q_{\text{real}})}{\partial x} = \theta_{\text{real}} - 2Q_{\text{real}} x - C'^T M \text{diag}(\gamma)(C'x - d') - C'^T (\frac{1}{4K}M + U)\gamma$$

(34)

Differentiating on both sides of the analytical solution of $x$, we get:

$$\frac{\partial x}{\partial \theta} = (2Q + C'^T M \text{diag}(\gamma) C')^{-1}$$

$$\frac{\partial x}{\partial Q} = (\theta + C'^T M \text{diag}(\gamma) d' - C'^T (U + \frac{M}{4K})\gamma)\frac{\partial (2Q + C'^T M \text{diag}(\gamma) C')^{-1}}{\partial Q}$$

(35)

To derive a simplified norm of $\frac{\partial x}{\partial Q}$, let $R = (2Q + C'^T M \text{diag}(\gamma) C')^{-1}$, $S = C'^T M \text{diag}(\gamma) C'$, $\beta = \theta + C'^T M \text{diag}(\gamma) d' - C'^T (U + \frac{M}{4K})\gamma$. With such notations, we can simplify the previous results to

$$x = (2Q + S)^{-1}\beta = R\beta, \quad \frac{\partial x}{\partial \theta} = R^T = R, \quad \frac{\partial f(x, \theta_{\text{real}}, Q_{\text{real}})}{\partial x} = \beta_{\text{real}} - R_{\text{real}}^{-1} x$$

(36)

and the derivative $\frac{\partial x_i}{\partial Q_{j,k}}$ and $\frac{\partial f(x, \theta_{\text{real}}, Q_{\text{real}})}{\partial Q_{j,k}}$ can be derived as follows:

$$\frac{\partial x_i}{\partial Q_{j,k}} = \sum_x \sum_y \frac{\partial R_{x,y}}{\partial Q_{j,k}} \frac{\partial x_i}{\partial R_{x,y}}$$

$$= \sum_y \frac{\partial R_{i,y}}{\partial Q_{j,k}} \beta_y$$

$$= \sum_y \beta_y \sum_p \sum_q \frac{\partial (2Q + S)_{p,q}}{Q_{j,k}} \frac{\partial R_{i,y}}{\partial (2Q + S)_{p,q}}$$

(37)

$$= \sum_y 2\beta_y \frac{\partial R_{i,y}}{\partial (2Q + S)_{j,k}}$$

$$= -\sum_y 2\beta_y R_{i,j} R_{k,y}$$

$$\frac{\partial f(x, \theta_{\text{real}}, Q_{\text{real}})}{\partial Q_{j,k}} = -\sum_i (\sum_y 2\beta_y R_{i,j} R_{k,y})(\beta_{\text{real},i} - (R_{\text{real}}^{-1} x)_i)$$

$$= 2\sum_i ((R_{\text{real}}^{-1} x)_i - \beta_{\text{real},i}) R_{i,j} \sum_y \beta_y R_{k,y}$$

(38)

Let $p_j = \sum_i R_{j,i}^T ((R_{\text{real}}^{-1} x)_i - \beta_{\text{real},i}), t_k = \sum_y \beta_y R_{k,y}$ ($p = R^T(R_{\text{real}}^{-1} x - \beta_{\text{real}}), t = x$), and finally

$$\frac{\partial f(x, \theta_{\text{real}}, Q_{\text{real}})}{\partial Q_{j,k}} = 2p_j x_k, \quad \frac{\partial f(x, \theta_{\text{real}}, Q_{\text{real}})}{\partial Q} = 2px^T$$

(39)

## D.3 Resource Provisioning

The problem formulation of resource provisioning is

$$\min_x \alpha_1^T \max(Cx - d, 0) + \alpha_2^T \max(d - Cx, 0), \ s.t. \ x^T \mathbf{1} = 1, x \geq 0 \tag{40}$$

Let $C' \begin{bmatrix} C \\ -C \\ -I \\ \mathbf{1}_{1\times \mathbf{n}} \\ -\mathbf{1}_{1\times \mathbf{n}} \end{bmatrix}$, $d' = \begin{bmatrix} d \\ -d \\ 0 \\ 1 \\ -1 \end{bmatrix}$, $\gamma = \begin{bmatrix} \alpha_1 \\ \alpha_2 \\ O(n^{1.5}E)\mathbf{1} \\ O(n^{1.5}E) \\ O(n^{1.5}E) \end{bmatrix}$, $P = (C'^T M \mathrm{diag}(\gamma) C')^{-1}$, $\beta =$

$(C'^T M \mathrm{diag}(\gamma) d' - C'^T(\frac{M}{4K} + U)\gamma)$, $\eta = M \mathrm{diag}(\gamma) d' - (\frac{M}{4K} + U)\gamma$. Then for the derivative $\frac{\partial f(x, C'_{\mathrm{real}})}{\partial x}$ and the analytical solution of $x$, we have

$$\frac{\partial f(x, C'_{\mathrm{real}})}{\partial x} = -C'^T_{\mathrm{real}} M_{\mathrm{real}} \mathrm{diag}(\gamma)(C'_{\mathrm{real}} x - d') - C'^T_{\mathrm{real}}(\frac{1}{4K} M_{\mathrm{real}} + U_{\mathrm{real}})\gamma$$

$$x = (C'^T M \mathrm{diag}(\gamma) C')^{-1}(C'^T M \mathrm{diag}(\gamma) d' - C'^T(\frac{M}{4K} + U)\gamma) = P\beta \tag{41}$$

According to the analytical solution of optimal point $x$, the derivative $\frac{\partial x_i}{\partial C'_{k,l}}$ is

$$\frac{\partial x_i}{\partial C'_{k,l}} = \sum_j \frac{\partial P_{i,j}}{\partial C'_{k,l}} \beta_j + \sum_j P_{i,j} \frac{\partial \beta_j}{\partial C'_{k,l}} \tag{42}$$

For the first term of the derivative above, we have:

$$\frac{\partial P_{i,j}}{\partial C'_{k,l}} = \frac{\partial (C'^T M \mathrm{diag}(\gamma) C')^{-1}_{i,j}}{C'_{k,l}}$$

$$= \sum_p \sum_q \frac{\partial (C'^T M \mathrm{diag}(\gamma) C')_{p,q}}{\partial C'_{k,l}} \frac{P_{i,j}}{P^{-1}_{p,q}}$$

$$= \sum_p \sum_q \frac{\partial \sum_x \sum_y C'_{x,p} M \mathrm{diag}(\gamma)_{x,y} C'_{y,q}}{\partial C'_{k,l}}(-P_{i,p}P_{q,j})$$

$$= \sum_q \sum_y [p == l] M \mathrm{diag}(\gamma)_{k,y} C'_{y,q}(-P_{i,p}P_{q,j}) + \sum_p \sum_x [q == l] M \mathrm{diag}(\gamma)_{x,k} C'^T_{p,x}(-P_{i,p}P_{q,j})$$

$$= \sum_q -P_{i,l}(M \mathrm{diag}(\gamma))_{k,*} C'_{*,q} P_{q,j} + \sum_p -P_{l,j} C'^T_{p,*}(M \mathrm{diag}(\gamma))_{*,k} P_{i,p}$$

$$= -(P_{i,l}(M \mathrm{diag}(\gamma))_{k,*} C' P_{*,j} + P_{l,j} P_{i,*} C'^T(M \mathrm{diag}(\gamma))_{*,k})$$

$$= -(P_{i,l}(M \mathrm{diag}(\gamma) C' P)_{k,j} + P_{l,j}(P C'^T M(\mathrm{diag}(\gamma)))_{i,k}) \tag{43}$$

Therefore, the simplified result for the first term of the derivative in Equation 42 is:

$$\sum_j \frac{\partial P_{i,j}}{\partial C'_{k,l}} \beta_j = -((M \mathrm{diag}(\gamma) C' P \beta)_k P_{i,l} + (P\beta)_l (P C'^T M \mathrm{diag}(\gamma))_{i,k}) \tag{44}$$

For the second term, we have:

$$\frac{\partial \beta_j}{\partial C'_{k,l}} = \frac{\partial (C'^T \eta)_j}{C'_{k,l}} = \frac{\partial \sum_p C'_{p,j} \eta_p}{\partial C'_{k,l}} = \eta_k[j == l] \tag{45}$$

The simplified second term is thus

$$\sum_j P_{i,j} \frac{\partial \beta_j}{\partial C'_{k,l}} = P_{i,l} \eta_k \tag{46}$$

Finally, the derivative can be written as

$$\frac{\partial x_i}{\partial C'_{k,l}} = -((M \mathrm{diag}(\gamma) C' P \beta)_k P_{i,l} + (P\beta)_l (P C'^T M \mathrm{diag}(\gamma))_{i,k}) + P_{i,l} \eta_k \tag{47}$$

# E Benchmark Details : Dataset and Problem Settings

Our code is public in the repo: https://github.com/PredOptwithSoftConstraint/PredOptwithSoftConstraint.

## E.1 Synthetic Linear Programming

### E.1.1 Prediction Dataset

We generate the synthetic dataset $\{\xi_i, \theta_i\}_{i=1}^N$ under a general structural causal model ([21]). In the original form, it is like:

$$
\begin{aligned}
z &\sim N(0, \Sigma) \\
\xi &= g(z) + \epsilon_1 \\
\theta &= h(z) + \epsilon_2
\end{aligned}
\tag{48}
$$

where $z$ is the latent variable, $\xi$ observed features of $z$, and $\theta$ the result variable caused by $z$. According to physical knowledge, $h$ can be a process of linear, quadratic, or bi-linear form. However, it is difficult to get an explicit form of reasonable $g$. Instead, $g^{-1}$ can be well-represented by deep neural networks.

Thus, alternatively, we use the following generative model:

$$
\begin{aligned}
\xi^* &\sim N(0, \Sigma) \\
z &= m(\xi^*) \\
\theta &= h(z) + \epsilon_2 \\
\xi &= \xi^* + \epsilon_1
\end{aligned}
\tag{49}
$$

where $m$ behaves as $g^{-1}$. In our experiment settings, $\Sigma = I + QQ^T$, where each element of $Q$ is generated randomly at uniform from $(0, 1)$. We set $m(x) = \sin(2\pi x B)$, where $B$ is a matrix whose elements are generated randomly at uniform in $\{0, 1\}$, and $\sin$ is applied element-wisely. We implement $h(z)$ as a MLP with two hidden layers, and the output is normalized to $(0, 1]$ for each dimension through different data points. Finally, we add a noise of $0.01\epsilon_x$ to $x$, where $\epsilon_x \sim N(0, 1)$; and $0.01\epsilon_\theta$ to $\theta$, where $\epsilon_\theta$ follows a truncated normal distribution which truncates a normal distribution $N(0, 1)$ to $[0, 1.5]$.

The dataset is split into training, validation and test sets with the proportions $50\%, 25\%, 25\%$ in respect. The batch size is set to 10 for $N = 100$, 50 for $N = 1000$, and 125 for $N = 5000$.

### E.1.2 Problem Settings

We generate hard constraint $Ax \leq b$, and soft constraint $Cx \leq d$. Each element of $A$ or $C$ is first generated randomly at uniform within $(0, 1)$, then set to 0 with probability of 0.5. We generate $b, d$ as $b = 0.5A\mathbf{1}$ and $d = 0.25C\mathbf{1}$. The soft constraint coefficient $\alpha$ is generated randomly at uniform from $(0, 0.2)$ for each dimension.

## E.2 Portfolio Optimization

### E.2.1 Dataset

The prediction dataset is daily price data of SP500 from 2004 to 2017 downloaded by Quandl API [24] with the same settings in [3]. Most settings are aligned with those in [3], including dataset configuration, prediction model, learning rate (initial 0.01 with scheduler), optimizer (Adam), gradient clip (0.01), the number of training epochs (20), and the problem instance size (the number of equities being $\{50, 100, 150, 200, 250\}$).

### E.2.2 Problem Settings

We set the number of soft constraints to $0.4$ times of $n$, where $n$ is the number of candidate equities. For the soft constraint $\alpha^T \max(Cx - d, 0)$, $\alpha = \frac{15}{n}v$, where each element of $v$ is generated randomly at uniform from $(0, 1)$; the elements of matrix $C$ are generated independently from $\{0, 1\}$, where the probability of 0 is 0.9 and 1 is 0.1. $K$ is set as 100.

### E.3 Resource Provisioning

#### E.3.1 Dataset

The ERCOT energy dataset [25] contains hourly data of energy output from 2013 to 2018 with 52535 data points. We use the first $70\%$ as the training set, the middle $10\%$ as the validation set, and the last $20\%$ as the test set. We normalize the dataset by dividing the labels by $10^4$, which makes the typical label becomes value around $(0.1, 1)$. We train our model with a batch size of 256; each epoch contains 144 batches. We aim to predict the matrix $C \in \mathbb{R}^{24 \times 8}$, where 24 represents the following 24 hours and 8 represents the 8 regions, which are {COAST, EAST, FWEST, NCENT, NORTH, SCENT, SOUTH, WEST}. The data is drawn from the dataset of the corresponding region. the decision variable $x$ is 8-dimensional, and $d = 0.5\mathbf{1} + 0.1N(0, 1)$.

#### E.3.2 Problem Settings

We test five sets of $(\alpha_1, \alpha_2)$, which are $(50 \times \mathbf{1}, 0.5 \times \mathbf{1})$, $(5 \times \mathbf{1}, 0.5 \times \mathbf{1})$, $(\mathbf{1}, \mathbf{1})$, $(0.5 \times \mathbf{1}, 5 \times \mathbf{1})$, and $(0.5 \times \mathbf{1}, 50 \times \mathbf{1})$, against two-stage with $L1$-loss, $L2$-loss, and weighted $L1$-loss. We use AdaGrad as optimizer with learning rate $0.01$, and clip the gradient with norm $0.01$. The feature is a $(8 \times 24 \times 77)$-dimensional vector for each matrix $C$; we adopted McElwee's Blog [26] for the feature generation and the model. Weighted L1-loss has the following objective:

$$\alpha_2^T \max(Cx - d, 0) + \alpha_1^T \max(d - Cx, 0) \tag{50}$$

Note that $\alpha_2$ and $\alpha_1$ are exchanged in the objective. Intuitively, this is because an under-estimation of entries of $C$ will cause a larger solution of $x$, which in turn makes $C_{\text{real}}x - d$ larger at test time, and vice versa. Another thing worth noting is that SPO+ cannot be applied to the prediction of the matrix $C$ or vector $d$, for it is designed for the scenario where the predicted parameters are in the objective.

## F  Supplementary Experiment Results

### F.1  Linear Programming with Soft Constraints

**The effect of $K$.** Table 4 provides the mean and standard deviation of regrets under varying values of $K$. we can see that our method performs better than all other methods with most settings of $K$.

Empirically, to find an optimal $K$ for a given problem and its experimental settings, a grid search with roughly adaptive steps suffices. For example,in our experiments, we used a proposal where neighbouring coefficient has $5x$ difference (e.g. $\{0.2, 1, 5, 25, 125\}$) works well. Quadratic objective, as in the second experiment, usually requires larger $K$ than linear objective as in the first and the third experiment.

| | | Regret | | | | |
|---|---|---|---|---|---|---|
| $N$ | Problem Size | ours($K = 0.2$) | $K = 1.0$ | $K = 5.0$ | $K = 25.0$ | $K = 125.0$ |
| 100 | (40, 40, 0) | 2.423±0.305 | 2.378±0.293 | 2.265±0.238 | 2.301±0.325 | **2.258±0.311** |
| | (40, 40, 20) | 2.530±0.275 | 2.574±0.254 | 2.384±0.277 | 2.321±0.272 | **2.350±0.263** |
| | (80, 80, 0) | **5.200±0.506** | 5.529±0.307 | 5.610±0.351 | 5.563±0.265 | 5.502±0.353 |
| | (80, 80, 40) | **4.570±0.390** | 4.747±0.470 | 4.758±0.531 | 4.780±0.450 | 4.796±0.624 |
| 1000 | (40, 40, 0) | 1.379±0.134 | 1.387±0.130 | 1.365±0.147 | **1.346±0.144** | 1.352±0.140 |
| | (40, 40, 20) | 1.587±0.112 | 1.553±0.114 | 1.523±0.109 | 1.507±0.102 | **1.506±0.102** |
| | (80, 80, 0) | 3.617±0.125 | **3.431±0.100** | 3.540±0.105 | 3.513±0.097 | 3.507±0.117 |
| | (80, 80, 40) | **2.781±0.165** | 2.817±0.167 | 2.819±0.155 | 2.820±0.157 | 2.826±0.197 |
| 5000 | (40, 40, 0) | 1.041±0.094 | 1.065±0.098 | 1.045±0.099 | 1.038±0.102 | **1.037±0.100** |
| | (40, 40, 20) | 1.278±0.072 | 1.248±0.076 | 1.223±0.075 | **1.220±0.071** | 1.221±0.074 |
| | (80, 80, 0) | 3.074±0.098 | **2.845±0.064** | 2.884±0.100 | 2.889±0.083 | 2.864±0.086 |
| | (80, 80, 40) | 2.217±0.123 | 2.222±0.111 | **2.172±0.098** | 2.174±0.113 | 2.177±0.105 |

Table 4: The mean and standard deviation of the regret of all $K$ in our experiment.

**Prediction error.** Table 5 and Table 6 are the mean and standard deviation of prediction MSE for all methods. We can find that two-stage methods have an advantage over end-to-end methods on

| | | MSE of $\theta$ | | | | |
|---|---|---|---|---|---|---|
| $N$ | Problem Size | L1 | L2 | SPO+ | DF | ours ($K = 0.2$) |
| 100 | (40, 40, 0) | **0.062±0.006** | 0.063±0.008 | 0.079±0.150 | 14.592±9.145 | 0.094±0.039 |
| | (40, 40, 20) | 0.061±0.005 | 0.062±0.006 | 0.071±0.006 | 6.414±4.451 | **0.061±0.007** |
| | (80, 80, 0) | **0.058±0.004** | 0.059±0.004 | 0.066±0.011 | 28.768±13.90 | 0.247±0.009 |
| | (80, 80, 40) | **0.059±0.004** | 0.059±0.004 | 0.085±0.008 | 8.98± 9.99 | 0.061±0.005 |
| 1000 | (40, 40, 0) | 0.031±0.001 | **0.031±0.001** | 0.034±0.002 | 34.931±14.309 | 0.036±0.003 |
| | (40, 40, 20) | **0.031±0.001** | 0.031±0.001 | 0.033±0.001 | 9.255±8.497 | 0.033±0.001 |
| | (80, 80, 0) | 0.030±0.001 | **0.029±0.001** | 0.032±0.002 | 90.590±35.298 | 0.222±0.009 |
| | (80, 80, 40) | 0.030±0.001 | **0.029±0.001** | 0.033±0.001 | 1.060±2.395 | 0.033±0.002 |
| 5000 | (40, 40, 0) | **0.022±0.000** | 0.022±0.000 | 0.025±0.001 | 72.083±46.685 | 0.025±0.001 |
| | (40, 40, 20) | 0.022±0.000 | **0.022±0.000** | 0.238±0.001 | 235.789±842.939 | 0.243±0.001 |
| | (80, 80, 0) | 0.021±0.000 | **0.021±0.000** | 0.023±0.001 | 147.144±51.101 | 0.196±0.011 |
| | (80, 80, 40) | 0.020±0.000 | **0.020±0.000** | 0.022±0.001 | 0.372±0.311 | 0.024±0.001 |

Table 5: The mean and standard deviation of the MSE loss of the predicted variable $\theta$ in the synthetic experiment on the test set. The two-stage methods have better performance on the accuracy of prediction.

| | | MSE of $\theta$ | | | |
|---|---|---|---|---|---|
| $N$ | Problem Size | ours($K = 1.0$) | ours($K = 5.0$) | ours($K = 25.0$) | ours($K = 125.0$) |
| 100 | (40, 40, 0) | 0.068±0.008 | 0.091±0.037 | 0.188±0.089 | 0.204±0.079 |
| | (40, 40, 20) | 0.094±0.031 | 0.204±0.074 | 0.244±0.105 | 0.343±0.200 |
| | (80, 80, 0) | 0.129±0.049 | 0.257±0.182 | 0.641±0.476 | 0.816±0.351 |
| | (80, 80, 40) | 0.094±0.013 | 0.151±0.035 | 0.172±0.091 | 0.177±0.042 |
| 1000 | (40, 40, 0) | 0.036±0.002 | 0.056±0.010 | 0.143±0.048 | 0.195±0.095 |
| | (40, 40, 20) | 0.060±0.010 | 0.127±0.051 | 0.172±0.071 | 0.193±0.078 |
| | (80, 80, 0) | 0.114±0.070 | 0.220±0.097 | 1.124±0.341 | 1.407±0.439 |
| | (80, 80, 40) | 0.039±0.002 | 0.060±0.006 | 0.078±0.010 | 0.080±0.008 |
| 5000 | (40, 40, 0) | 0.026±0.001 | 0.033±0.004 | 0.076±0.025 | 0.089±0.037 |
| | (40, 40, 20) | 0.036±0.004 | 0.073±0.019 | 0.110±0.038 | 0.113±0.026 |
| | (80, 80, 0) | 0.152±0.079 | 0.097±0.040 | 0.703±0.177 | 0.895±0.205 |
| | (80, 80, 40) | 0.026±0.001 | 0.032±0.003 | 0.038±0.004 | 0.040±0.007 |

Table 6: The mean and standard deviation of the MSE loss of the predicted variable $\theta$ in the synthetic experiment for different values of $K$.

prediction error. however, as demonstrated in our experiments, such advantage does not necessarily lead to the advantage on the regret performance. Surprisingly, although DF[9] works generally on par with other baselines in regret performance, it generates a very large MSE error compared with our method; closer examination suggests that there are some outlier cases where the prediction loss is magnitudes higher than others. Also, the prediction error of DF gets much higher when no soft constraint exists, which is probably because the optimal solution remains the same when all parameters to predict are scaled by a constant factor.

**Detailed behaviour of SPO+.** As mentioned in the main paper, SPO+ quickly becomes overfited in our experiments, shown by Table 7 as empirical evidence. In our experiments, early stopping starts at epoch 8, and maximum number of run epochs is 40; the results show that SPO+ stops much earlier than other methods under most settings. Figure 8 plots the timely comparison of test performance during training, where SPO+ get overfited quickly, although it performs better in the earlier period of training.

**Statistical significance tests.** Table 9 shows the results of significance test (one-tailed paired $t$-test), with the assumption that the means of two distributions are the same. The results show that our method is significantly better than other methods under almost all settings.

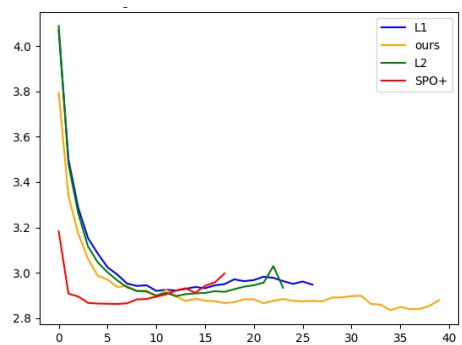

Figure 8: The average regret on test set of $K = 25.0$, training set size $5000$ and problem size $(80, 80, 0)$ with respect to the number of epochs (methods other than ours does not reach $40$ in this figure, for all runs are early-stopped before epoch $40$). While SPO+ is better than two-stage in the best performance, it overfits rather quickly.

| $N$ | Problem Size | L1 | L2 | SPO+ | DF | ours ($K = 0.2$) |
|---|---|---|---|---|---|---|
| | | | | Average Episode | | |
| 100 | (40, 40, 0) | 20.46±10.20 | 12.53±3.00 | 19.6±8.58 | 29.53±10.24 | 20.53±10.39 |
| | (40, 40, 20) | 21.47±9.58 | 14.13±4.47 | 17.8±9.98 | 25.46±11.41 | 22.47±10.29 |
| | (80, 80, 0) | 22.6±5.78 | 22.6±8.52 | 20.07±7.23 | 30.87±8.83 | 24.53±8.02 |
| | (80, 80, 40) | 25.2±8.38 | 24.73±8.88 | 19.33±764 | 24±9.06 | 28.33±8.89 |
| 1000 | (40, 40, 0) | 14.67±2.61 | 13.2±1.74 | 10±2.04 | 27.93±7.42 | 13.6±2.53 |
| | (40, 40, 20) | 13.2±2.37 | 13.4±2.50 | 10.33±1.40 | 24.6±8.14 | 10.67±1.23 |
| | (80, 80, 0) | 19.07±2.25 | 18.07±1.83 | 11.87±2.00 | 25.2±6.41 | 24.27±11.02 |
| | (80, 80, 40) | 19.93±2.66 | 19±1.31 | 12.33±1.95 | 25.4±9.22 | 17.73±4.08 |
| 5000 | (40, 40, 0) | 16.47±2.59 | 16.27±1.33 | 11.87±2.10 | 23.33±7.34 | 11.8±2.60 |
| | (40, 40, 20) | 17.13±2.59 | 15.27±1.79 | 12.8±2.88 | 21.67±7.95 | 13.4±2.77 |
| | (80, 80, 0) | 20.67±3.20 | 20.8±2.18 | 13.7±2.16 | 29.33±7.22 | 30.07±9.48 |
| | (80, 80, 40) | 20.47±2.64 | 21±2.88 | 12.2±1.61 | 17.6±6.84 | 18.53±5.26 |

Table 7: The mean and standard deviation of the number of epochs run in each instance (capped at $40$). Note that SPO+ is significantly more prone to overfitting in our experiment settings.

### F.2 Portfolio Optimization

**Statistical significance tests.** Table 10 shows the result of significance test (one-tailed paired $t$-test), with the assumption that the means of two distributions are the same. The results show that our method is significantly better than other methods under every setting.

### F.3 Resource Provisioning

**Statistical significance tests.** Table 11 shows the result of significance test (one-tailed paired $t$-test), with the assumption that the means of two distributions are the same.

## G  Computing Infrastructure

All experiments are conducted on Linux Ubuntu 18.04 bionic servers with 256G memory and 1.2T disk space with no GPU, for GPU does not suit well with Gurobi.[10] Each server has 32 CPUs, which are Intel Xeon Platinum 8272CL @ 2.60GHz.

For the first experiment, we use a few minutes to get one set of data [11] with training set size $100$, and about $4 - 5$ hours to get one set of data with training set size $5000$. For the second experiment, we use

---

[10] see Gurobi's official support website: https://support.gurobi.com/hc/en-us/articles/360012237852-Does-Gurobi-support-GPUs-

[11] Running all methods simultaneously under one particular parameter setting.

|  |  | $t$-value | | | |
| --- | --- | --- | --- | --- | --- |
| $N$ | Problem Size | L1 | L2 | SPO+ | DF |
| 100 | (40, 40, 0) | 3.884 | 4.615 | 5.194 | 1.614 |
|  | (40, 40, 20) | 5.173 | 5.248 | 6.836 | 1.569 |
|  | (80, 80, 0) | 6.024 | 5.661 | 4.523 | 3.610 |
|  | (80, 80, 40) | 2.308 | 1.724 | 3.622 | 1.936 |
| 1000 | (40, 40, 0) | 8.438 | 5.271 | 4.860 | 1.115 |
|  | (40, 40, 20) | 6.547 | 6.212 | 8.562 | 2.620 |
|  | (80, 80, 0) | 12.620 | 10.925 | 7.979 | 2.751 |
|  | (80, 80, 40) | 7.116 | 4.465 | 6.223 | 8.230 |
| 5000 | (40, 40, 0) | 7.130 | 7.808 | 7.546 | 1.175 |
|  | (40, 40, 20) | 10.410 | 10.238 | 6.555 | 2.382 |
|  | (80, 80, 0) | 8.361 | 7.502 | 6.874 | 0.869 |
|  | (80, 80, 40) | 6.550 | 4.388 | 5.066 | 11.620 |

Table 8: The $t$-value of the one-tailed paired $t$-test between all other methods and our methods under optimal $K$.

|  |  | $p$-value | | | |
| --- | --- | --- | --- | --- | --- |
| $N$ | Problem Size | L1 | L2 | SPO+ | DF |
| 100 | (40, 40, 0) | $8.272 \times 10^{-4}$ | $2.006 \times 10^{-4}$ | $6.809 \times 10^{-5}$ | 0.058 |
|  | (40, 40, 20) | $7.076 \times 10^{-5}$ | $6.162 \times 10^{-5}$ | $4.054 \times 10^{-6}$ | 0.064 |
|  | (80, 80, 0) | $1.561 \times 10^{-5}$ | $2.936 \times 10^{-5}$ | $2.388 \times 10^{-4}$ | $5 \times 10^{-4}$ |
|  | (80, 80, 40) | 0.018 | 0.053 | 0.001 | 0.032 |
| 1000 | (40, 40, 0) | $3.662 \times 10^{-7}$ | $5.917 \times 10^{-5}$ | $1.263 \times 10^{-4}$ | 0.137 |
|  | (40, 40, 20) | $6.489 \times 10^{-6}$ | $1.133 \times 10^{-5}$ | $3.078 \times 10^{-7}$ | 0.007 |
|  | (80, 80, 0) | $2.444 \times 10^{-9}$ | $1.544 \times 10^{-8}$ | $7.064 \times 10^{-7}$ | 0.005 |
|  | (80, 80, 40) | $2.600 \times 10^{-6}$ | $2.667 \times 10^{-4}$ | $1.114 \times 10^{-5}$ | $2.937 \times 10^{-9}$ |
| 5000 | (40, 40, 0) | $2.545 \times 10^{-6}$ | $9.074 \times 10^{-7}$ | $1.342 \times 10^{-6}$ | 0.125 |
|  | (40, 40, 20) | $2.834 \times 10^{-8}$ | $3.487 \times 10^{-8}$ | $6.406 \times 10^{-6}$ | 0.012 |
|  | (80, 80, 0) | $4.082 \times 10^{-7}$ | $1.435 \times 10^{-6}$ | $3.818 \times 10^{-6}$ | 0.196 |
|  | (80, 80, 40) | $6.460 \times 10^{-6}$ | $3.097 \times 10^{-4}$ | $8.606 \times 10^{-5}$ | $1.585 \times 10^{-12}$ |

Table 9: The $p$-value of the one-tailed paired $t$-test between all other methods and our methods under optimal $K$.

about an hour to get one set of data with $N = 50$, and $2 - 3$ days to get one set of data with $N = 250$. For the third experiment, we use around $6 - 7$ hours to obtain one set of data. Though our method is slower to train than two-stage methods, it is 2-3x faster to train than KKT-based decision-focused method.

| | $t$-value | | | $p$-value | | |
|---|---|---|---|---|---|---|
| #Equities | L1 | L2 | DF | L1 | L2 | DF |
| 50 | 10.329 | 10.628 | 6.141 | $3.213\times10^{-8}$ | $2.186\times10^{-8}$ | $1.279\times10^{-5}$ |
| 100 | 21.414 | 23.995 | 6.200 | $2.125\times10^{-12}$ | $4.495\times10^{-13}$ | $1.157\times10^{-5}$ |
| 150 | 13.402 | 14.503 | 5.690 | $1.119\times10^{-9}$ | $3.971\times10^{-10}$ | $2.789\times10^{-5}$ |
| 200 | 13.617 | 13.104 | 4.329 | $9.086\times10^{-10}$ | $1.500\times10^{-9}$ | $3.465\times10^{-4}$ |
| 250 | 13.904 | 14.483 | 3.205 | $6.915\times10^{-10}$ | $4.044\times10^{-10}$ | 0.003 |

Table 10: The $t$-value and $p$-value of the one-tailed paired $t$-test between all other methods and our methods.

| | $t$-value | | | $p$-value | | |
|---|---|---|---|---|---|---|
| $\alpha_1/\alpha_2$ | L1 | L2 | Weighted L1 | L1 | L2 | Weighted L1 |
| 100 | 13.133 | 9.357 | 6.429 | $2.914\times10^{-9}$ | $2.115\times10^{-7}$ | $1.576\times10^{-5}$ |
| 10 | 12.484 | 14.231 | 10.897 | $5.622\times10^{-9}$ | $1.493\times10^{-6}$ | $3.190\times10^{-8}$ |
| 0.1 | 1.184 | 5.073 | 11.001 | 0.127 | $8.486\times10^{-5}$ | $1.416\times10^{-8}$ |
| 0.01 | 0.535 | 1.998 | 5.083 | 0.300 | 0.032 | $8.33\times10^{-5}$ |

Table 11: The $t$-value ($p$-value) of the one-tailed paired $t$-test between all other methods and our methods.