# OpenReview forum: "A Surrogate Objective Framework for Prediction+Programming with Soft Constraints"
_NeurIPS.cc/2021/Conference — NeurIPS 2021 Poster_

### Official Review · Reviewer_FE8x · 2021-07-13

**Rating:** 4
**Confidence:** 3

**Summary:**

This paper presents an approach for "predict, then optimize" problems that is amenable to derivative-based learning methods. It involves converting hard constraints into soft constraints (i.e. implicitly enforced through objective penalization). The main theoretical contributions are bounds on the magnitude of objective change that can be attributed to infeasibility with respect to the original hard constraints.

**Ethical Concerns:**

None.

**Limitations And Societal Impact:**

I do not feel that the authors adequately confront the limitations of their work. Most of the results restrict the problem data to be nonnegative. They claim that "constraint parameters usually represent weights, quantities or their thresholds, and thus are always nonnegative." This is overly broad--there are many, many "realistic" optimization problems that do not satisfy this condition. The wide swath of network optimization problems are just one set of examples. This restriction is a reasonable enough one as many problems _do_ satisfy it, but the paper should be suitably rewritten to soften its claims.

There is no direct negative societal impact of the work.

**Main Review:**

The paper is quite difficult to read. The theoretical results are presented informally in the body of the paper to the point of confusion, and it is difficult to glean what the conclusions from them should be. As such, it is difficult to assess the technical soundness of the work. As one example, the second condition on H (line 137) is written confusingly and informally (what does "increments" mean in this context? How do you formally measure "gain" with respect to violation of the linear constraints?). As a second, the statement of Lemma 1 is also informal--what is the maximin discussed taken with respect to? As a third, Corollary 4 does not formally define what "cardinal constraints" means (should be "cardinality constraints"?) and what their implications are for the structure of the problem data. As a fourth, the theoretical results in Section 3 only seem to indicate asymptotic guidance as to the maximum penalty weights to be considered, not how they should actually be set in practice, let alone to guarantee optimality w.r.t. the true problem. This vagueness continues in Section 4 and the Appendix, where (contra 3(b) in the checklist) the penalty terms gamma are described only in asymptotic notation; the reader is not told how these values are actually set in the experiments.

The approach presented to smooth the max constraints is not particularly surprising, though I have no reason to question its novelty. The computational results seem promising, though the lack of details about the implementation make it difficult to see how readers would glean insight into how to take these results and apply to new problems or extend them.

**Time Spent Reviewing:**

1.5

---

> ### Author Response · Authors · 2021-08-10
> **Response to Reviewer FE8x**
>
> We thank the reviewer for the valuable feedback on our work. We first apologize for the difficulty of understanding. To help the reviewer understand our work well, we now briefly introduce the paper's organizational structure again, and clarify the concerned notation, statements, and potential limitation.
>
> Concern 1: About difficulty to follow with the theoretical parts.
>
> Response: We will briefly introduce the organization of theoretical parts as follows.
>
> - The problem formulation is given in Equation 1, with 3 specific applications (Equations 2 $\sim$ 4) which are later solved and evaluated in the experimental section.
>
> - The methodology includes (1) the differentiable surrogate function of soft constraint $max(z, 0)$ as Equation 7 and its derivatives as Equation 8 in section 3.2, and (2) transformation of hard constraints into soft constraints as Equation 6 at the end of Section 3.1.3.
>
> - Theories in Section 3.1 support the transformation of hard constraints into soft ones -- Theorem 2, Theorem 3 and Theorem 6 guarantee the feasibility of transforming $Ax \leq b$, $x \geq 0$, and $Bx=c$, respectively. Besides, Lemma 1 is the foundation to prove most of other theorems; Corollary 4 and Theorem 5 are for special forms of constraints.
>
> We would like to note that for the inactivity, we described the theorems in an informal way in the main body, while rewritten them formally with proofs in Appendix.
>
> Concern 2: About concerns on notation and statements.
>
> Response: We explain the reviewer's concerns with respect to notions and statements in details.
>
> - "Increments" on line 137 is the same thing as "gain" formally defined on line 155 and denoted by $R$;
>
> - The formal definition of "maximin" is Eq. 12 in Appendix A, between Line 560 and Line 561 (we did not state the formal definition in our main content due to page limit);
>
> - Cardinal constraints should be cardinality constraints, which is explained on line 178 ("where the entries of $A$ are binary") and they imply a set of "decision for selection" problems (for example, a relaxation of the discrete optimization problem with ``selecting at most $k$ items from a particular subset" constraint; the second and third experiment of this paper are also the instances, where the decision maker determines a ratio distributed over different items);
>
> - The penalty term $\gamma$ is set either to a non-tuned reasonable constant (e.g. 20 to 100) or constant (e.g. around $5$) times $||\theta||\sqrt{n}$,  which can be respectively checked in Line 81 and Line 133 at synthetic\_linear\_programming/util.py of our source code (the URL of the source code is listed in the footnote of page 27). We believe that interested readers may find insights on the implementation by taking a look at our source code. Of course, we will also update the value of parameter $\gamma$ in the paper accordingly.
>
> We appreciate the reviewer's advice on the presentation very much, and we will surely improve the presentation of the next draft according to the suggestions listed by the reviewer. However, we would like to note that the technical soundness itself is not hindered by the current presentation: (1) As confirmed by Reviewers WmBF and rNgo, the methodological approach is generally well-supported with clear and overall rigorous proofs; (2) Moreover, our method has already examined to be working well on real-life data (Experiments 2 and 3).
>
> Concern 3: About the discussion of potential limitation.
>
> Response: As for the potential limitation, we will discuss more in the next draft. This paper mainly focuses on the scope where the constraint parameters mainly represent weights and quantities, and thus requires the non-negativity (or box constraints under special cases) of the constraint matrix. In the future, we will develop theoretical guarantees for more general cases, and the claims will be properly adjusted as the reviewer suggests in our next draft. Practically, though, our method does not necessarily require the non-negativity of $A$; this potential limitation serves purely for proving the upper-bound of penalty coefficient.

---

> > ### Comment · Reviewer_FE8x · 2021-08-18
> > **Concern 2**
> >
> > > "Increments" on line 137 is the same thing as "gain" formally defined on line 155 and denoted by R;
> >
> > If the concepts are identical, please use the same technical terminology to make this clear.
> >
> > > The formal definition of "maximin" is Eq. 12 in Appendix A, between Line 560 and Line 561 (we did not state the formal definition in our main content due to page limit);
> >
> > Even if informal, the paper and its results should be unambiguous and comprehensible on its own, with the appendix supplementing with additional details and content. I would argue that this is not currently the case.
> >
> > > Cardinal constraints should be cardinality constraints, which is explained on line 178 ("where the entries of  are binary")...
> >
> > I am confused: The "entries of A being binary" does not coincide with cardinality constraints as commonly understood, so I am not sure how line 178 adequately explains the concept, and so my concern stands.
> >
> > > The penalty term gamma is set either to a non-tuned reasonable constant...
> >
> > Please include this discussion in the paper/appendix.

---

> > > ### Author Response · Authors · 2021-08-18
> > > **Thanks for the constructive comments**
> > >
> > > > "Increments" on line 137 is the same thing as "gain" formally defined on line 155 and denoted by R;
> > > >> If the concepts are identical, please use the same technical terminology to make this clear.
> > >
> > > Good idea. We will move the definition to the place it firstly appears.
> > >
> > > > The formal definition of "maximin" is Eq. 12 in Appendix A, between Line 560 and Line 561 (we did not state the formal definition in our main content due to page limit);
> > > >> Even if informal, the paper and its results should be unambiguous and comprehensible on its own, with the appendix supplementing with additional details and content. I would argue that this is not currently the case.
> > >
> > > Agree, we will revise all theorems in the main paper to make them formal and self-contained, with the same statements in the Appendix. By the way, the formula of "maximin" is already given in the body of Lemma 1 (Line 546-547 in Appendix A), and Line 560~561 further derives an equivalent form which is used in the later proofs of left theorems.
> > >
> > >
> > > > Cardinal constraints should be cardinality constraints, which is explained on line 178 ("where the entries of are binary")...
> > > >> I am confused: The "entries of A being binary" does not coincide with cardinality constraints as commonly understood, so I am not sure how line 178 adequately explains the concept, and so my concern stands.
> > >
> > > You are correct. The "cardinality constraints" is not part of our target optimization/programming problem. We just use the "cardinality" of $A$'s row vector $A_{,j}$, i.e., $||A_{,j}||_0$, as a bound to finish the proof. We will eliminate such unnecessary usage.
> > >
> > >
> > > > The penalty term gamma is set either to a non-tuned reasonable constant...
> > > >> Please include this discussion in the paper/appendix.
> > >
> > > Agree, we will do it in the revision.

---

### Official Review · Reviewer_WmBF · 2021-07-13

**Rating:** 7
**Confidence:** 2

**Summary:**

The paper proposes a new method to integrate soft constraints in the predict+programming paradigm that focuses on mathematical programming models with a prediction component that needs to be considered during optimization/solving.
The new method works by reformulating the problem into piecewise linear constraints and then optimizing a surrogate for the original problem.
A theoretical analysis and motivation is provided as well as an experimental evaluation that underlines the effectiveness of the method.

**Limitations And Societal Impact:**

There is no dedicated discussion of societal impacts, but it is also not particularly relevant in this work.

**Main Review:**

As a disclaimer, I'm broadly familiar with predict+optimize, but am not a specialist in this area of mathematical programming and did not follow the theory in detail.
However, the methodological approach seems sound and well supported by the theory.

I was wondering why the methodology was framed as "prediction+programming" rather the, in my view, more established "predict+optimize". Also, even though mathematical programming is an established term, using "programming" might lead to some confusion in regards to software programming.

In general, I think this is solid work and of relevance for this area.

**Time Spent Reviewing:**

3

---

> ### Author Response · Authors · 2021-08-10
> **Response to Reviewer WmBF**
>
> We thank the reviewer for the appreciation of our work.
>
> Concern 1: About the term usage of 'programming' vs 'optimization'.
>
> Response: We chose the word 'programming' here to show that the scope of our work is focused on the particular set of mathematical optimization (i.e., linear and quadratic programming problems) with real-life applications; however, we agree with the reviewer about the potential misunderstandings caused by current name, and will take the helpful advice in the next draft.

---

### Official Review · Reviewer_rNgo · 2021-07-16

**Rating:** 7
**Confidence:** 3

**Summary:**

The paper develops a surrogate function with closed-form solution for objective functions with "soft constraints," here defined as those with the term $\alpha^T \\max\\{0,z(x)\\}$ for affine $z$, component-wise $\\max$, and $\alpha > 0$. The goal is to apply such functions within the context of smart "predict-then-optimize" (SPO) by also relaxing the hard constraints with a sufficiently large penalty $\alpha$. To develop their surrogate, the authors evaluate bounds on the utility gained when violating the soft constraints using primarly a geometric perspective, specifically evaluating the angles or corners of infeasible points with respect to the convex hull of feasible solutions (and related properties). Numerical results compare their approach with the standard SPO+ loss function and a DF metric for portfolio optimization.

**Limitations And Societal Impact:**

Not discussed in the text. The paper could greatly benefit from a discussion concerning the limitations and trade-offs of their methodology.

**Main Review:**

**Originality**

The paper is part of a current stream of papers that investigate different strategies to smooth the underlying loss function within the SPO framework, which is itself a quite challenging problem. Within that context, I found the polyhedral perspective provided by this paper quite interesting and, to the best of my knowledge, original as well. The general ideas are quite relevant given that $\\max\\{0,z\\}$ functions are quite pervasive, e.g., in A.I. and operations.


**Quality**

Overall, I believe the paper is well structured but I do have some few technical concerns. More specifically:

1. When softening hard constraints and calculating their penalty, the terms involve angle, smallest corner calculations, and other complex functions such as $\\sin$. I wonder if there could be challenges associated with numerical stability? Furthermore, how time consuming was calculating such angles and auxiliary parameters for the instances tested?

2. Many of the bounds are defined as within function classes $O(\\cdot)$. From my understanding of the structural results, the actual functions to be used in the penalties could be associated with large constants (e.g., due to $c_1$, l.651, in pg.18, supplemental material), which could make the approximation too weak. Perhaps I misunderstood, but I believe some further discussion on how to prescribe the bounds precisely could be beneficial here, including some intuition on the quality of these bounds.


**Clarity**

The main text is well-written and the supplemental material is comprehensive. My few comments here are as follows.

3. It would be best if the assumptions are more clearly described in the text. In particular, non-negativity of $\alpha$ could not necessarily be always the case (e.g., in Benders or cut-generation linear programs, where similar ideas could be useful). Furthermore, I believe the constant $E$ is an important factor for the development of the results and it could be emphasized further.

4. The proofs were overall rigorously written and are clear to the best of my analysis. The auxiliary figures are extremely helpful. However, Theorems 3 and 5 are relatively more difficult because many statements are posed more vaguely (e.g., in 591-595, how to project the cone to its tip?). I would suggest just to extend them somewhat with more hand-holding and appropriate formalism, if possible.

5. Some statements in the main require references (or at least authors could cite related papers earlier) - for example: "modern solvers ... can already efficiently find optimal for most large scale programming problems," or "it may model overhead of under-provisioning, over-provisioning of goods."

**Significance**

In my view, this is a paper that provides a clear and useful contribution and may attract a broader interest within the A.I. community. The idea of "softening" constraints by exploiting polyhedral corners/angles is original within the context of SPO, to the best of my knowledge. Nonetheless, the calculation of the softening constraint is more involved and some practical aspects (quality of the bounds, complexity, and numerical stability) requires some more clarification, which I believe can be easily addressed by the authors.

**Minor Points**

- The references have a few errors. For example, there is no journal information in [10], reverence [7] should be updated with CPLEX v20 (2021) etc.
- "more loyal" --> "closer?"
- There is often a missing "the" prior to the word "programming" throughout the text.


**Time Spent Reviewing:**

6

---

> ### Author Response · Authors · 2021-08-10
> **Response to Reviewer rNgo**
>
> We thank the reviewer for the appreciation of our work. Our responses to the concerns are listed as follows.
>
> Concern 1: About technical concerns in quality.
>
> Response: We would like to mention that the theoretical results are for extreme cases, and real-life data are more lenient. In practice, for example, our guidance is simply setting the penalty coefficients $\alpha$ using a reasonable constant (e.g. 20 to 100) or constant (e.g. around 5) times $||\theta||\sqrt{n}$ instead of actually calculating the angles, which has been implemented in our source code (see line 81 and 133 at synthetic\_linear\_programming/util.py; the URL of the source code is listed in the footnote of page 27).
>
>
> Concern 2: About presentation and supplementary materials.
>
> Response: We appreciate the reviewer for your suggestion on the constant in the asymptotic bound; we will give analysis on the constant in the next draft. As for clarity, significance and other minor points, we will follow the constructive advice and update them respectively.

---

> > ### Comment · Reviewer_rNgo · 2021-08-23
> > **Concern**
> >
> > Thanks for your feedback. I am somewhat concerned now that the constant was set to a fixed value and not calculated explicitly. Couldn't that be an additional source of errors? How could one determine those constants in general (e.g., is the "5x the bound..." described in the feedback something general to consider?). It would be great if the authors could clarify those points.

---

> > > ### Author Response · Authors · 2021-08-24
> > > **response to concerns on setting coefficients of softened constraints**
> > >
> > > We think there should be no additional error sources as long as the original hard constraints are satisfied when the solving process converges. Note that we only set the part of the coefficients for softening original hard constraints, while maintains the part ($\alpha$ in Eq.1) for original soft constraints.
> > >
> > > As for a general rule of thumb in determining the constants, take eq.2 in the paper as an example. We advice the user to start from a reasonable constant or a constant times $\sqrt{n}$, where such "reasonable constant" is the product of a constant factor (e.g. 5-10) and a roughly estimated upper bound of $\theta$ (which corresponds to E in our bounds) with specific problem settings; then alternately increase the constant and time an extra $\sqrt{n}$ while resetting the constant until the program converges with original hard constraints satisfied. In our experiments, we hardly observe situations where such process goes for two or more steps.

---

### Official Review · Reviewer_Vxh4 · 2021-07-18

**Rating:** 6
**Confidence:** 4

**Summary:**

This paper provides a method for addressing the "prediction + programming" problem in the case where the programming problem contains a $\max(z,0)$ term in the objective. To do so, the authors rewrite the optimization problem as an unconstrained optimization problem by converting all hard constraints of the optimization problem to soft constraints in the objective. They then pick the coefficients of those soft constraints in a way that is meant to prevent constraint violations, and then relax these soft constraints via a surrogate function. They then embed the resultant approximate optimization problem within a method where some $\theta$ is output by a neural network, the approximate optimization problem is solved for that value of $\theta$, and gradients are updated by differentiating end-to-end through the approximate optimization problem. The authors demonstrate their method on three synthetic settings, and report improved performance over two-stage methods and SPO+.

**Limitations And Societal Impact:**

As mentioned in my main review, I think the authors could do a better job flagging some of the assumptions they make and describing their significance.

**Main Review:**

The paper addresses an important prediction + programming setting, and provides an interesting surrogate objective framework for addressing it. However, one of the central motivational claims of the paper is flawed, which unfortunately compromises the premise of the paper. In additional, several of the assumptions made throughout are rather strong, and this is not properly flagged.

### Central motivational claim

The central claim on which this work is built is that existing methods cannot handle the setting where there is a $\max(z,0)$ term in the objective function. However, this setting can indeed be handled by existing methods. For instance, the class of programming problems studied can be differentiated through using cvxpylayers (see example code at the end of this review), and KKT differentiation approaches such as those proposed in OptNet [13] can also be used (by obtaining the optimal $z^\star$ via some solver, rewriting $\max(z^\star, 0) = z^\star$ or 0 depending on the value of this optimal solution, and differentiating through the KKT conditions of the resultant problem). These differentiable layers can then be directly used within the end-to-end learning approaches proposed in [9, 11] to address the prediction + programming problem.

Now, the approaches I have described above are certainly far from perfect: For instance, they are likely to be computationally expensive to train (as KKT differentiation-based approaches tend to be). In contrast, the method proposed in the present paper is potentially cheaper to train. However, this is not the axis of improvement that the paper currently aims to demonstrate (i.e., time comparisons to cvxpylayer- or OptNet-based methods are not shown).

If the authors were to revamp the paper (both the motivation and the experiments) to address the problem of decreasing computational cost (rather than the current motivation of addressing a previously-unaddressed setting), I think this would significantly strengthen the paper (and accordingly, I would potentially be open to increasing my score).

### Assumptions

Some of the assumptions made throughout are also rather strong. While this is okay for the purposes of developing a specialized method, these should be properly flagged and described as limitations rather than being described as common circumstances:
* Line 74: The assumption that all terms in $A$ are non-negative is rather strong, in particular because encoding box constraints on $x$ (that is, $x_l \leq x \leq x_u$ for $x_l, x_u > 0$) would require the presence of negative terms in $A$. Such box constraints are rather common.
* Line 141: The assumption that the l2 norm of $\nabla f$ is bounded is also potentially rather strong, particularly in the quadratic case (Equation 3). This should be flagged and discussed.
* Line 193: It would be helpful if the definition of $P$ could be more clearly described, as it is not immediately clear from the description in the main text why $\lambda_{max}$ would be bounded in practice.

### Other comments

* Line 149: Is "active constraints at $x$" supposed to refer to all constraints that are violated by $x$? (Given that $x \not\in \mathcal{C}$, "active constraints" may not be the correct terminology.)
* Line 201: This statement should be corrected. As described above, Equation 5 can indeed be differentiated through at its optimum since the "max" operation can be replaced by either the linear term or 0 depending on the value of $x^\star$ that is obtained. This rewritten problem can then be differentiated through via the implicit function theorem.

### Sample cvxpylayers code

As mentioned above, here is some sample code for setting up a differentiable layer for Equation 2 via cvxpylayers.

	import cvxpy as cp
	import torch
	from cvxpylayers.torch import CvxpyLayer
	import numpy as np

	# Random seeds for param generation (picked so the problem is feasible)
	np.random.seed(0)
	torch.random.manual_seed(6)

	# Initialize problem vars and params
	n, m, k = 2, 3, 5
	x = cp.Variable(n)
	theta = np.random.randn(n)
	# pick alpha as the ones vector (see "sum" in the objective)
	A = cp.Parameter((m, n))
	b = cp.Parameter(m)
	B = cp.Parameter((n, n))
	c = cp.Parameter(n)
	C = cp.Parameter((k, n))
	d = cp.Parameter(k)

	# Set up cvxpy problem
	objective = cp.Maximize(theta@x - cp.sum(cp.maximum(C@x + d, 0)))
	constraints = [A@x <= b, B@x == c, x >= 0]
	problem = cp.Problem(objective, constraints)
	assert problem.is_dpp()  # True

	# Create cvxpylayer
	cvxpylayer = CvxpyLayer(problem, parameters=[A, b, B, c, C, d], variables=[x])
	A_tch = torch.randn(m, n, requires_grad=True)
	b_tch = torch.randn(m, requires_grad=True)
	B_tch = torch.eye(n, requires_grad=True)
	c_tch = torch.ones(n, requires_grad=True)
	C_tch = torch.randn(k, n, requires_grad=True)
	d_tch = torch.randn(k, requires_grad=True)

	# Solve the problem
	solution, = cvxpylayer(A_tch, b_tch, B_tch, c_tch, C_tch, d_tch)

	# Compute the gradient of the sum of the solution with respect to problem params
	solution.sum().backward()   # works

**Time Spent Reviewing:**

3

---

> ### Author Response · Authors · 2021-08-10
> **Response to the concern about the assumptions and necessity to flag them properly**
>
> We agree with the reviewer that such assumptions need more explicit discussion on availability, as there are some applications that require negative terms in $A$ (such as network optimization problem mentioned by the reviewer FE8x) or where $\nabla f$ are unbounded (especially for quadratic programming problems with open feasible domain). However, the real-world problems that satisfy our assumption are quite common, ranging from bipartite matching, knapsack problems, logistic optimizations such as transport/production planning, portfolio optimization (Experiment 2 in our paper) to cloud source/energy provisioning (Experiment 3 in our paper). Also, the example raised by the reviewer with respect to line 74 can be mitigated by substitution techniques (e.g. assign new $x'=x-x_l$), aligning the box to the origin; then for box constraints with $x\geq 0$ as its corner, we have both theorem 5 and 6 for theoretical guarantee on penalty coefficients. We will take the reviewer's advice, though, and add more explicit discussion on the availability of our method in the next draft.
>
> Besides, we explain other concerns raised by the review as follows.
> - Line 193: We put the clarification of $P$ in appendix, at the proof of theorem 6 due to page limit. We will update our main content accordingly in the next draft.
>
> - Line 149: Yes, it is supposed to refer to all constraints that are violated by $x$.

---

> > ### Comment · Reviewer_Vxh4 · 2021-08-24
> > **Good response, but some remaining points**
> >
> > Thank you to the authors for their detailed response. In particular, I really appreciate the authors running experiments showing that the cvxpylayers implementation does not work out of the box for the LP setting. The clarification about some of the assumptions also makes sense. For that reason, I have increased my score from 3 to 5.
> >
> > However, I am still leaning towards rejection, notably because I think there are "easy" baselines to which the authors are not comparing.
> >
> > ### Baselines
> >
> > **Quadratic programming setting:** In the setting where $g(\theta, x)$ is quadratic, it would be possible to:
> > * Solve the optimization problem in the forward pass using a standard solver (e.g., cvxpy, since the problem is DPP)
> > * After obtaining the solution, in the backward pass, rewrite the objective as $g(\theta, x^\star) − \alpha^T z^\star$ if $z^\star > 0$, else as $g(\theta, x^\star)$.
> > * Use KKT differentiation to differentiate through the rewritten problem.
> >
> > I'll note again that this approach is likely to be very slow, as KKT differentiation approaches are. In addition, if $z^\star = 0$, the gradients may be incorrect. However, I would expect this approach to work correctly in most cases, as well as address more general problems (e.g., no bounds on $\nabla f$).
> >
> > Relatedly, it's not clear to me whether the failures experienced by the cvxpylayers solver in the tests the authors ran is due to known difficulties in differentiating through LPs, or whether the issue is indeed due to the ReLU term in the objective. It would be illustrative to see the results on the QP case.
> >
> > **Linear programming setting:** I agree that the singularity of KKT conditions in LP-style settings ("singularities in cases such as max and linear+max objectives") is an open problem, and outside the scope of this particular paper to address as part of the theory. However, I don't think it is unreasonable to ask for appropriate baseline comparisons in the experiments (notably, those using regularization approaches), given that regularization approaches have indeed been proposed in previous work (as the authors point out). As above, this would entail solving the optimization problem in the forward pass using a standard solver, and then rewriting the problem + using KKT differentiation on the rewritten problem in the backward pass.
> >
> > ### Clarity
> >
> > Separately, I do agree with Reviewer FE8x that the clarity of the submission does need to be improved, notably that some of the algorithmic components were difficult to follow. That said, I trust the authors will act on the feedback they received, so this does not affect my score.

---

> > > ### Author Response · Authors · 2021-08-25
> > > **explanation on baselines**
> > >
> > > Thank you for the appreciation! We admit the importance of comparing with 'easier' baselines when there exist. Our related work, already done by the original paper or being carried during this discussion, are as follows.
> > >
> > > - **Quadratic programming**. As suggested by the reviewer, the KKT-based method could work in this case. Actually in the original paper, we have already adopted such a baseline for comparison, namely *DF* in the second application. The method description is as follows: (line 277) "To make DF work ... we use local 2nd-order Taylor expansion as a quick fix", where the 2nd-order Taylor expansion is calculated by first solving the optimization problem with a solver and then determining the coefficient of linear terms. The related code is the function `train_portfolio` in *portfolio_optimization/portfolio_utils.py* (note: the repo is given in the footnote of page 27 in the complementary material), especially line 308 to 331, where (line 320) `C = gamma @ C0` does the same work suggested by the reviewer. In this application, the DF method shows better performance than two-stage methods, meanwhile is not as good as ours. This implies that the known difficulty of differentiation through the LPs is at least a major reason for divergence in our test.
> > >
> > > - **Linear programming**. We agree that it is reasonable to add a KKT-based baseline here, given the fact that by rewriting the LP-style objective with an additional L2 regularization term, the KKT-based differentiation works right in many practice. After completing all experiments, we will report the result in this thread and in the next draft.

---

> > > > ### Comment · Reviewer_Vxh4 · 2021-08-25
> > > > **further discussion on baselines**
> > > >
> > > > Thank you for clarifying, and for the code/appendix pointers. Looking through the code, it seems that the authors are indeed taking the approach I suggested via the DF baseline in the quadratic programming setting. My apologies for missing that.
> > > >
> > > > The authors' plan to include this baseline for the LP case also sounds good.
> > > >
> > > > Based on these points, I have further increased my score.
> > > >
> > > > I would ask in the final version, however, the authors please do make some of the details clearer in the main body of the paper, as the description of e.g. the portfolio optimization problem is not self contained (since the problem definition is not in the main text, phrases like "we use local 2nd-order Taylor expansion as a quick fix" are not clear without digging deep into the code and appendix). Space could be made for this additional description by moving the final form of the derivative (Equation 9) to the appendix; the problem setup and salient "tricks" are likely more important to describe in the main body. (Similar comment for the "Resource Provisioning" section.)

---

> > > > > ### Author Response · Authors · 2021-08-31
> > > > > **further response and baseline experimental results**
> > > > >
> > > > > Thank you for your constructive advice! We will update our paper accordingly in the final version to clarify all details of problem settings for both our method and baselines.
> > > > >
> > > > > The KKT-based method for linear programming setting has been complemented. As discussed before, we rewrite the objective with an additional L2-regularization (with coefficient 0.1 as in Wilder et al.'s[1,2] code).  The results are summarized as follows.
> > > > >
> > > > > | N | Problem Size | Regret |
> > > > > | ---- | ---------------| --------|
> > > > > | 100 | (40, 40, 0) | 2.478(+/-0.425) |
> > > > > |        | (40, 40, 20) | 2.536(+/-0.376) |
> > > > > |        | (80, 80, 0) | 5.756(+/-0.317) |
> > > > > |        | (80, 80, 40) | 4.902(+/-0.537) |
> > > > > | 1000 | (40, 40, 0) | 1.434(+/-0.268) |
> > > > > |         | (40, 40, 20) | 1.529(+/-0.151) |
> > > > > |        | (80, 80, 0) | 3.532(+/-0.102) |
> > > > > |        | (80, 80, 40) | 3.351(+/-0.212) |
> > > > > | 5000 | (40, 40, 0) | 1.078(+/-0.092) |
> > > > > |         | (40, 40, 20) | 1.291(+/-0.091) |
> > > > > |        | (80, 80, 0) | 2.869(+/-0.085) |
> > > > > |        | (80, 80, 40) | 2.748(+/-0.165) |
> > > > >
> > > > > Compared to Table 1 in our original paper, KKT-based method is generally on par with other baselines, while worse than our method. The code for the additional experiments is already updated at our repo.
> > > > >
> > > > > [1] Melding the Data-Decisions Pipeline: Decision-Focused Learning for Combinatorial Optimization, in AAAI 2019
> > > > >
> > > > > [2] Automatically Learning Compact Quality-aware Surrogates for Optimization Problems, in NeurIPS 2020

---

> ### Author Response · Authors · 2021-08-10
> **Response to the concern about the central motivation claim**
>
> We thank the reviewer for the valuable opinion and constructive advice on paper improvement.
>
> We think that cvxpylayers cannot handle the $max(z, 0)$ directly, and through our experiments the sample code given by the reviewer just cannot give the right gradients of parameters.
>
> - Experimental studies. According to the review, in order to study the effectiveness of the sample cvxpylayers code (which is originally given in the review) in our scenarios, we have conducted the following experiments: We modified the sample cvxpylayers code to solve the problem of Equation 2, a simplified version of our first experiment. Our experimental results demonstrate that the predict+optimize process of cvxpylayers code fails on test set, as shown in Table 1. On the other hand, our method and others in our paper get close to optimal objective value in few epochs. The full code extended with data generator and complete solving process is presented in the end. The code is self-contained, and was tested under the newest version of environment (torch 1.9.0, cvxpy 1.1.13, cvxpylayers 0.1.5 with Python 3.8).
>
> **Table 1**: Test Performance (Optimal objective value: 8.268)
> -- - ----- - -----
> Epoch | Objective Value | Regret
> --| ----- | -----
> 1 | 2.192 | 6.076
> 2 | 1.992 | 6.275
> 3 | 1.991 | 6.277
> 4 | 2.046 | 6.221
> 5 | 2.137 | 6.131
> 6 | 2.112 | 6.155
> 7 | 2.094 | 6.173
> 8 | 2.094 | 6.174
> 9 | 2.002 | 6.266
> 10 | 1.995 | 6.272
> 11 | 1.995 | 6.273
> 12 | 2.010 | 6.257
> 13 | 2.002 | 6.266
> 14 | 2.014 | 6.253
> 15 | 2.008 | 6.260
> 16 | 2.014 | 6.254
> 17 | 1.965 | 6.303
> 18 | 1.942 | 6.325
> 19 | 1.950 | 6.317
> 20 | 1.944 | 6.323
>
> - Informal theoretical analysis. According to [3][4], for $max(z, 0)$'s convex curvature, it would possibly be incorporated into an existing disciplined convex programming (DCP) problem while keeping the disciplined parametrized programming (DPP) properties, and thus the problem with $max$ operator might pass the DPP check and be ``handled" by the cvxpylayers. However, it can also be easily verified, by manually deriving the differentiation of the the optimality conditions, that there may exist the singularities in cases such as max and linear+max objectives. Also, the same issue exists in Karush–Kuhn–Tucker (KKT) differentiation method developed in OptNet. To fix this issue, extra efforts are required to ensure regularities of the optimality conditions. This issue in pure linear cases was also identified by a recent work ([1], in Section 7.2).
>
> Nevertheless, we believe with extra efforts, the existing approaches of differentiating KKT and homogeneous self-duality (HSD) have the potential to handle the $max(z, 0)$ in principle. Such efforts would include extending the objective by adding an regularization term for specific problem, as in the pure linear objective ([1][2]). We would like to note that exploring such efforts is beyond the scope of this paper, and we will study them in our future work.
>
> In the next draft, we will update our statement to precisely reflect the progress of existing methods. Note that this motivational claim is not the basis on which our method is built. In fact, our proposed surrogate approach is an alternative of the optimality differentiation approach, and shows advantage on tested problems with max in the objective.
>
> **References**:
>
> [1] End-to-End Constrained Optimization Learning: A Survey, in IJCAI 2021.
>
> [2] Melding the data-decisions pipeline: Decision-focused learning for combinatorial optimization, in AAAI 2019.
>
> [3] CVXPY doc, https://www.cvxpy.org/tutorial/dcp/.
>
> [4] Differentiable Convex Optimization Layers, in NeurIPS 2019.
>
>
> ```python
> import cvxpy as cp
> import numpy as np
> import torch
> import torch.nn as nn
> from torch.optim import Adagrad
> from sklearn.model_selection import train_test_split
> from tqdm import tqdm
> import math
> from cvxpylayers.torch import CvxpyLayer
> import gurobipy as gp
> from gurobipy import GRB
>
> seed = 7355611
> np.random.seed(seed)
> torch.manual_seed(seed)
>
> feature_size, hidden_size, decision_size = 32, 64, 16
> data_size = 1500
> epoch, batch_size = 5, 30
> num_soft_constr, num_hard_constr = 10, 10
>
> net = nn.Sequential(
>     nn.Linear(feature_size, hidden_size),
>     nn.ReLU(),
>     nn.Linear(hidden_size, decision_size)
> ).double()
>
> optimizer = Adagrad(net.parameters())
>
> # data generation
> P, q = np.random.random((decision_size, feature_size)), np.random.random((decision_size, 1)) # y = Px + q
> x = np.random.random((feature_size, data_size)) - 0.5
> y = np.matmul(P, x) + q
> x, y = (x + 0.01 * np.random.normal()).T, (y + 0.01 * np.random.normal()).T
> xi_train, xi_test, theta_train, theta_test = train_test_split(x, y, test_size=0.2, random_state=seed)
>
> # constraint generation
> def SparseUniformMatrix(shape, low, high, sparsity=0.5):
>     x = torch.empty(shape)
>     x = nn.init.uniform_(x, low, high)
>     rows, cols = x.shape
>     num_zeros = int(math.ceil(sparsity * rows))
>     for col_idx in range(cols):
>         row_indices = torch.randperm(rows)
>         zero_indices = row_indices[:num_zeros]
>         x[zero_indices, col_idx] = 0
>     return x
>
> A = SparseUniformMatrix(shape=(num_hard_constr, decision_size), low=0, high=1, sparsity=0.5)
> C = SparseUniformMatrix(shape=(num_soft_constr, decision_size), low=0, high=1, sparsity=0.5)
> b = A @ torch.ones((decision_size, 1)) * 0.5
> d = C @ torch.ones((decision_size, 1)) * 0.25
> alpha = nn.init.uniform_(torch.empty(num_soft_constr, 1), 0, 0.2).numpy()
> alpha0, A0, b0, C0, d0 = alpha, A.numpy(), b.numpy(), C.numpy(), d.numpy()
>
> # For evaluation
> def getval(theta, x, alpha0, A0, b0, C0, d0):
>     return theta.T.dot(x) - alpha0.T.dot(np.maximum(np.matmul(C0, x.reshape(-1, 1)) - d0, 0))
>
> def getopt(theta, alpha0, A0, b0, C0, d0): # get optimal true value.
>     x0 = np.zeros(A0.shape[1])
>     ev = gp.Env(empty=True)
>     ev.setParam('OutputFlag', 0)
>     ev.start()
>     m = gp.Model("matrix1", env=ev)
>     x = m.addMVar(shape=theta.shape[0], vtype=GRB.CONTINUOUS, name='x')
>     z = m.addMVar(shape=d0.shape[0], vtype=GRB.CONTINUOUS, name='z')
>     m.setObjective(theta.T @ x - alpha0.T @ z, GRB.MAXIMIZE)
>     m.addConstr(z >= 0, name="c1")
>     m.addConstr(z >= C0 @ x - d0.squeeze(), name='c2')
>     m.addConstr(x >= 0, name="c3")
>     m.addConstr(A0 @ x <= b0.squeeze(), name='c4')
>     m.optimize()
>     return getval(theta, x.getAttr('x'), alpha0, A0, b0, C0, d0), x.getAttr('x')
>
> class PerfCompare:
>     def __init__(self, alpha, A, b, C, d):
>         self.alpha, self.A, self.b, self.C, self.d = alpha, A, b, C, d
>
>     def compare(self, theta_pred, theta, backlog=None):
>         # reward metrics
>         optvals = np.zeros(theta.shape[0])
>         for i in range(optvals.shape[0]):
>             x = getopt(theta[i].reshape(-1, 1), self.alpha, self.A, self.b, self.C, self.d)[1]
>             optvals[i] = getval(theta[i].reshape(-1, 1), x.reshape(-1, 1), self.alpha, self.A, self.b, self.C, self.d)
>         vals = np.zeros(theta_pred.shape[0])
>         for i in range(vals.shape[0]):
>             x1 = getopt(theta_pred[i].reshape(-1, 1), self.alpha, self.A, self.b, self.C, self.d)[1]
>             vals[i] = getval(theta[i].reshape(-1, 1), x1.reshape(-1, 1), self.alpha, self.A, self.b, self.C, self.d)
>         avg_optval = np.mean(optvals)
>         avg_val = np.mean(vals)
>         avg_reg = np.mean(optvals - vals)  # mean of regrets
>         if backlog is not None:
>             backlog.write(str(avg_optval) + " " + str(avg_val) + " " + str(avg_reg) + "\n")
>             backlog.flush()
>         return {'avg_optval': avg_optval, 'avg_val': avg_val, 'avg_reg': avg_reg}
>
> if __name__ == "__main__":
>     des = open("DF_size_" + str(decision_size) + "_" + str(num_soft_constr) + "_" + str(num_hard_constr) + "_datasize_" + str(data_size) + "seed" + str(seed) + ".txt", "w")
>     des2 = open("DF_size_" + str(decision_size) + "_" + str(num_soft_constr) + "_" + str(num_hard_constr) + "_datasize_" + str(data_size) + "seed" + str(seed)+ "test.txt", "w")
>     pc = PerfCompare(alpha0, A0, b0, C0, d0)
>
>     for i in tqdm(range(epoch)):
>         indices = np.random.permutation(xi_train.shape[0])  # shuffle
>         batches = math.ceil(indices.shape[0] / batch_size)
>         for bidx in range(batches):
>             bs = min(batch_size, indices.shape[0] - batch_size * bidx)
>             st = bidx
>             idx = indices[batch_size * st:  batch_size * st + bs]
>             xi_batch = xi_train[idx]
>             theta_batch = theta_train[idx]
>
>             theta_batch_pred = net(torch.from_numpy(xi_batch))
>
>             loss = torch.zeros(1)
>             for j in range(bs):
>                 theta, theta_true, alpha, A, b, C, d = theta_batch_pred[j], torch.from_numpy(theta_batch[j]), alpha0, A0, b0, C0, d0
>                 x_var = cp.Variable(C.shape[1])
>                 theta_para = cp.Parameter(C.shape[1])
>                 constraints = [A @ x_var <= b.reshape(-1), x_var >= 0]
>                 objective = cp.Maximize(theta_para @ x_var - alpha.T @ cp.maximum(C @ x_var - d.reshape(-1), 0))
>                 problem = cp.Problem(objective, constraints)
>                 cvxpylayer = CvxpyLayer(problem, parameters=[theta_para], variables=[x_var])
>                 x, = cvxpylayer(theta)
>                 obj = theta_true @ x - torch.from_numpy(alpha.T).double() @ torch.maximum(
>                         torch.from_numpy(C).double() @ x.view(-1, 1) - torch.from_numpy(d).double(),
>                         torch.zeros(C.shape[0], 1).double())
>                 loss += -obj.view(1)
>             loss /= bs
>
>             optimizer.zero_grad()
>             loss.backward()
>             nn.utils.clip_grad_norm_(net.parameters(), 0.001)
>             optimizer.step()
>
>             results = pc.compare(theta_batch_pred.detach().cpu().numpy(), theta_batch, des)
>
>             if bidx % batches == batches - 1:
>                 theta_test_pred = net(torch.from_numpy(xi_test))
>                 results_test = pc.compare(theta_test_pred.detach().cpu().numpy(), theta_test, des2)
> ```

---

### Decision · Program_Chairs · 2021-09-27

**Decision:**

Accept (Poster)

**Comment:**

A majority of reviewers recommend acceptance although, notably, a majority of reviewers also raise clarity concerns with the paper.  For the most part, the reviewers felt that these concerns were satisfactorily addressed in the significant rebuttal discussion.  The reviewers request the authors to revise the paper in light of the reviews and rebuttal discussion, which all reviewers agree would be manageable within the time frame for revision of accepted NeurIPS papers.